

# A novel network security situation assessment model based on multiple strategies whale optimization algorithm and bidirectional GRU

Shengcai Zhang[1], Qiming Fu[1], Dezhi An[1], Zhenxiang He[1] and Zhenyu Liu[2]

[1] School of Cyber Security, Gansu University of Political Science and Law, Lanzhou, Gansu Province, China
[2] School of Information Science and Engineering, Lanzhou University, Lanzhou, Gansu Province, China

## ABSTRACT

The rapid development of the internet has brought about a comprehensive transformation in human life. However, the challenges of cybersecurity are becoming increasingly severe, necessitating the implementation of effective security mechanisms. Cybersecurity situational awareness can effectively assess the network status, facilitating the formulation of better cybersecurity defense strategies. However, due to the low accuracy of existing situational assessment methods, situational assessment remains a challenge. In this study, a new situational assessment method, MSWOA-BiGRU, combining optimization algorithms and temporal neural networks, was proposed. Firstly, a scientific indicator system proposed in this research is used to calculate the values of each indicator. Then, the Analytic Hierarchy Process is used to derive the actual situation values, which serve as labels. Taking into account the temporal nature of network traffic, the BiGRU model is utilized for cybersecurity situational assessment. After integrating time-related features and network traffic characteristics, the situational assessment value is obtained. During the evaluation process, a whale optimization algorithm (MSWOA) improved with a mix of strategies proposed in this study was employed to optimize the model. The performance of the proposed MSWOA-BiGRU model was evaluated on publicly available real network security datasets. Experimental results indicate that compared to traditional optimization algorithms, the optimization performance of MSWOA has seen significant enhancement. Furthermore, MSWOA-BiGRU demonstrates superior performance in cybersecurity situational assessment compared to existing evaluation methods.

# INTRODUCTION

Artificial intelligence, cloud computing, and the Internet of Things are becoming more and more popular, integrating deeply into our lives. As these technologies develop, the scale and complexity of networks have continued to increase, and the current network

Corresponding author
Shengcai Zhang,
zsc6731@gsupl.edu.cn

environment has undergone unprecedented changes. As the number of internet-connected systems increases, the attack surface is also expanding, leading to greater attack risks. The techniques used by attackers are becoming increasingly advanced. Vulnerabilities are being exploited and it is increasingly more difficult to evade malicious software with security measures, allowing attacks to persist for extended periods without detection. A number of severe cybersecurity incidents have occurred across the world in recent years. This indicates that cybersecurity incidents caused by malicious attacks or disruptions are becoming more prevalent, and networks and information systems are facing a multitude of threats from network attacks. In response to the escalating network threat landscape, both domestic and international enterprises and researchers have been actively engaging in research related to cybersecurity. Currently, network security measures include installing antivirus and anti-spyware software (*Li & Wei, 2019*), implementing firewalls to block unauthorized network access (*Togay et al., 2022*), and utilizing intrusion detection and defense systems for threat identification (*Sarker et al., 2020*). However, as the information age and networks continue to evolve, data management on a large scale presents even greater challenges to traditional network environments. Existing security measures are unable to analyze network situations and maintain cybersecurity from a macroscopic perspective when faced with increasingly diverse and complex network security information data, thus failing to meet the demands of network security.

The ability to take effective and timely measures to understand the overall state of the network and respond has become an urgent and critical issue that needs to be addressed. Network situational awareness, as a proactive defense technology to ensure network security, is well-equipped to counter security threats in complex network environments. Situation awareness was first proposed by *Endsley (1988)*. This method provides a comprehensive consideration of the situation and its future development. In 1999, situational awareness was introduced into the field of network security and was first applied in the research of next-generation intrusion detection systems (*Bass, 1999*). Network situational awareness (*Cheng & Lang, 2012*) assesses the current state of the network based on environmental factors and predicts its future state in the near term. Network security situational assessment, as an important component of network situational awareness, can help to build effective models based on relevant security events to evaluate the overall threat level faced by the network system. It helps security administrators understand the security status of the entire network and supports the formulation of rational security response decisions. Therefore, conducting research on network security situational assessment is highly necessary and plays a proactive role in enhancing network security.

While traditional methods based on mathematical models and knowledge reasoning can integrate the overall network situation to provide decision recommendations for network administrators to some extent, they have become inadequate in the era of big data when it comes to handling a large volume of network traffic and attacks. Consequently, the accuracy of network security situation assessment is not particularly high. Deep neural networks (DNNs) present a potential solution to these issues due to their strong ability to represent high-dimensional complex data (*Liu et al., 2023*). Most existing works on

network security situation assessment primarily focus on current traffic features, which are more informative (*Gao et al., 2015*; *Liu et al., 2017*; *Shi & Chen, 2017*; *Chen, Yin & Sun, 2018*; *Qiang, Wang & Dang, 2018*; *Zhao & Liu, 2018*; *Han et al., 2019*; *Yang et al., 2020b, 2021*). Although the characteristics of current network traffic are undeniably important, in the real world, network traffic doesn't appear as independent entities but rather as a large volume of network traffic occurring in a time-series manner. The nature of network traffic is not solely dependent on current traffic features but also on the correlations between them (*Sharafaldin, Habibi Lashkari & Ghorbani, 2018*; *Tian, 2020*; *Lohrasbinasab et al., 2022*; *Yang et al., 2023*). We refer to this kind of correlation as temporal features, and this highlights the fact that focusing solely on current traffic features cannot effectively enhance the accuracy of situation assessment.

In order to enhance the accuracy of network security situation assessment, this study proposes MSWOA-BiGRU as a novel approach for network security situation assessment.

The main contributions of this study are summarized as follows:

(1) This study proposes a novel hybrid network security situational assessment method called MSWOA-BiGRU, which combines the new variant of the whale optimization algorithm-multiple strategies whale optimization algorithm (MSWOA) with the bidirectional gated recurrent unit (BiGRU) model. Both current traffic characteristics and timing characteristics can be considered, and MSWOA can find the best hyperparameter combination. This method effectively improves the accuracy of network security situational assessment.

(2) A new optimization algorithm called MSWOA is proposed. MSWOA is an improved version of the original optimization algorithm, WOA. It incorporates three enhancement strategies. Compared to traditional optimization algorithms, MSWOA exhibits higher optimization accuracy and faster convergence speed. Moreover, the whale optimization algorithm with multiple strategy improvements (MSWOA) demonstrates an outstanding capability to escape local optima.

(3) The proposed method was subjected to performance testing on the publicly available network security dataset, UNSW-NB15. The experimental results indicated that the new network security situational assessment method, MSWOA-BiGRU, exhibited small errors and a higher degree of curve fitting between the situational assessment values and the actual situational values. This method accurately evaluates the network security situation.

The remaining sections of this article are organized as follows. 'Related Works' introduces the related work. 'Construction of NSSA index system' describes the methodology for constructing the indicator system and calculating the actual values of the situation. The improved whale optimization algorithm based on multiple strategies' provides a detailed explanation of the new variant of the whale optimization algorithm, MSWOA, along with its performance testing. In the 'Proposed NSSA model and algorithm the newly proposed network security situational assessment method, MSWOA-BiGRU, is introduced. 'Datasets and Evaluation Criteria' describes the dataset and evaluation criteria. The experimental results and analysis of the network security situational assessment method are provided in 'Experiment and Analysis'. Finally, the conclusion is presented.

## RELATED WORKS

Network security situational assessment has become a widely researched area in academia and the industry, leading to diverse solutions. Generally, existing methods for network security situational assessment can be categorized into three types: those based on mathematical statistics, those based on knowledge reasoning, and those based on machine learning.

Early network security situational assessment models primarily adopted methods based on mathematical statistics. These models comprehensively considered factors influencing the network and achieved network security situational assessment by constructing evaluation functions. Mathematics-based methods include the analytic hierarchy process (AHP), the set pair analysis method, the fuzzy comprehensive evaluation method, and the multi-attribute utility function method. *Chen et al. (2006)* designed a hierarchical network security situational assessment method based on the analytic hierarchy process (AHP). They utilized AHP to construct a hierarchical network threat situational assessment indicator system to determine the weights of network threats. This marks the first application of AHP in network security situational assessment, providing researchers with an intuitive understanding of the security threat status for services, hosts, and local networks. The study enables a comprehensive view of the overall system's security condition. It facilitates the identification of security behaviors, the adjustment of security strategies, and the enhancement of system security performance. The model also offers technical references for subsequent researchers who employ mathematical statistical methods for network security situational assessment. Based on improved AHP and fuzzy comprehensive evaluation methods, a multilevel fuzzy comprehensive evaluation model is proposed to evaluate the network security situation by *Bian, Wang & Mao (2013)*. A fuzzy clustering algorithm fusion security situation assessment model for DDoS attacks is presented by *Zhang et al. (2018)*. Their model is reasonably efficient at assessing the cyber posture of DDoS attacks. The evaluation method based on mathematical statistics collects elements of network security situational awareness and comprehensively analyzes them, ultimately achieving situational assessment through constructed functions. Its advantage lies in the ability to consider multiple influencing factors, thereby forming a comprehensive assessment of network security situational awareness. However, mathematical models can be manipulated by expert experience, lacks objective standards, and is not conducive to long-term assessment.

In addition to the methods based on mathematical statistics, there is another approach in early network security situational assessment, known as the knowledge reasoning method. There are several methods that are available, including fuzzy reasoning, Bayesian networks, Markov processes, and the Dempster-Shafer (D-S) evidence theory. This type of method is mainly represented by fuzzy theory. The construction of the evaluation model is based on expert knowledge and an experience database, and security conditions are assessed through logical reasoning. *Sodiya & Oladunjoye (2007)* designed a threat modeling technique based on fuzzy theory. This method represents the first application of knowledge reasoning in network security situational assessment. The approach makes threat analysis easier and enables effective examination of potential threats in the network

system. *Zihao et al. (2017)* computes metrics representing network conditions at each layer and then fuses these metrics with D-S evidence theory to estimate influence. A hidden Markov model extension was used to assess the cyber security situation by *Liao et al. (2020)*. According to *Hu et al. (2020)*, the improved hidden Markov model (I-HMM) is proposed as a method to assess the cyber security risk and to determine the level of cyber security risk. In order to timely and accurately reflect security risk, the model uses a learning algorithm to set parameters. Although knowledge reasoning has performed well in network security situational assessment, in cases where the information is relatively simple, the limited availability of prior knowledge makes it difficult to conduct subsequent reasoning, leading to a decrease in the evaluation accuracy of knowledge reasoning methods. On the other hand, when dealing with highly complex information, the increase in quantification levels and the complexity of the calculation process reduce the assessment efficiency, making real-time evaluation impractical.

The emergence of machine learning methods has brought new technological support for network security situational assessment. *Chen, Yin & Sun (2018)* utilized the support vector machine (SVM) and the gravitational search algorithm to design a hybrid network security assessment method that combines optimization algorithms with machine learning methods. This approach achieved high evaluation accuracy. An improved cuckoo search (ICS) evolutionary algorithm was proposed by *Li et al. (2018)* as a method for pre-training a BPNN. This neural network is used to evaluate information security for IoT systems risk and to improve accuracy and stability. *Gao et al. (2015)* designed an SVM-based information system security risk assessment model and employed the artificial fish swarm algorithm (AFSA) for optimization. This approach utilized AFSA to optimize SVM, resulting in significantly higher accuracy and faster convergence speed. Machine learning-based evaluation methods assess network security situational awareness by building models. The advantages of such methods lie in their higher learning capabilities, enabling representation learning of a large volume of network data features. However, the downside is that the abundance of learned features results in complex computations and prolonged modeling time. This reduces the efficiency of evaluation, particularly in real-time demanding network environments.

In the age of the Internet of Things, network environments have become increasingly complex, and the massive growth of network traffic data presents significant challenges to network security situational assessment. In this context, cybersecurity researchers have recognized the immense advantages of deep learning methods in handling large-scale data. The application of deep learning algorithms is a trend in developing network security situational assessment. Deep learning, a branch of artificial intelligence, mimics the working principles of the human brain's neural networks. It involves the hierarchical stacking of neural network structures to learn and represent high-level features, enabling automated processing and analysis of complex data. Researchers are gradually integrating deep learning methods into the field of situational assessment. *Yang et al. (2020a)* proposed a network security situational assessment method based on variational autoencoders (VAEs), which effectively identifies network threats. *Lin et al. (2019)* demonstrated through threat detection experiments that compared to LSTM and gated

recurrent unit (GRU) models, bidirectional gated recurrent units (BiGRU) achieve higher accuracy in network security situational assessment. *Li & Zhao (2019)* applied the improved LSTM model to network security situational assessment, and the results demonstrated that the proposed method effectively understood and evaluated network security situations. Compared to traditional situational assessment methods, deep learning methods allow for effective analysis and the learning of large amounts of data, effectively detecting attack behaviors in network traffic data. As a result, it demonstrates stronger advantages in addressing the constantly evolving network environment. Choosing parameters in a neural network is a challenge. A good parameter selection not only enhances the model's robustness but also effectively improves its performance (*Lorenzo et al., 2017*; *Moosbauer et al., 2021*; *Vincent & Jidesh, 2023*).

To address this challenge, using optimization algorithms is a good choice. The Whale Optimization Algorithm (WOA) is a bio-inspired heuristic algorithm based on the group behavior of whales during hunting (*Hemasian-Etefagh & Safi-Esfahani, 2020*). In applications, WOA has been extensively used for real-world problems such as IoT node clustering (*Bozorgi et al., 2021*), PID controllers (*Saafan & El-Gendy, 2021*), online social networks (*Jain, Katarya & Sachdeva, 2020*), multi-threshold image segmentation (*Abd Elaziz, Lu & He, 2021*; *Anitha, Immanuel Alex Pandian & Akila Agnes, 2021*), aircraft trajectory optimization (*Zhang et al., 2020*), high-dimensional problems (*Zhang & Wen, 2021*), and more. With the rapid development of neural networks in recent years, the whale algorithm has also been widely applied for neural network parameter optimization (*Aljarah, Faris & Mirjalili, 2018*; *Kushwah, Kaushik & Chugh, 2021*). Moreover, some researchers have compared the performances of popular optimization algorithms including Whale Optimization Algorithm (WOA), Grey Wolf Optimizer (GWO), and Particle Swarm Optimization (PSO) on neural networks. The results indicate that WOA exhibits faster convergence speed and better optimization performance in finding global optimal solutions for neural networks, and can avoid local optima (*Gharehchopogh & Gholizadeh, 2019*). To enhance the performance of the algorithm, modifications and improvements to WOA are required (*Agrawal, Kaur & Sharma, 2020*; *Chen et al., 2020*; *Chen, Li & Yang, 2020*; *Sharmila & Vijayarani, 2021*; *Aala Kalananda & Komanapalli, 2021*). Given that the WOA algorithm shows better convergence speed and accuracy than other algorithms, this article adopts the Whale Optimization Algorithm as the foundational optimization method for model parameter optimization.

Overall, the emergence of deep learning algorithms provides new approaches and technological means for network security situational assessment, representing the development trend in this field.

# CONSTRUCTION OF NSSA INDEX SYSTEM

## Indicator selection

In network security situational assessment, the selection of indicators is an indispensable step in constructing the situational assessment indicator system. The chosen indicators directly reflect the evaluators' perspectives and ideas, and, to some extent, influence the situational assessment results and the applicability of the indicator system.

It is important to maintain the independence of the indicators while ensuring their comprehensiveness. Ensuring the independence of indicators means that each indicator holds unique informational value in the network security situational assessment. The independence among indicators ensures that they provide different perspectives and information without duplication or overlap. Each indicator should focus on different aspects or features to provide diversified and comprehensive security information. If there is redundancy or repetition among indicators, the accuracy of the network security situational assessment may be affected as the number of indicators increases, potentially leading to biased or inaccurate evaluation results.

Secondly, balancing the consideration of dynamic and static indicators is essential. Static indicators, such as the total number of hosts and the importance of various devices, reflect the network system's fundamental performance and configuration information over a certain period, contributing to a more objective analysis of the network's basic security status. These static indicators provide an evaluation of the network system's fundamental configuration and performance, which aids in identifying security vulnerabilities and weaknesses. On the other hand, dynamic indicators exert continuous influence on the network's condition through their own dynamic changes, rather than being limited to specific moments or nodes. By continuously monitoring the network's state in real-time, dynamic indicators can help uncover potential security issues and threats. This ongoing monitoring allows for the timely detection of network security incidents and abnormal behaviors, enabling security teams to promptly implement corresponding response measures. Therefore, the integration of dynamic and static indicators is crucial for a better and more comprehensive reflection of the overall network security status.

Thirdly, scientific principles must be followed and must consider the actual network environment. Appropriate indicators should be selected based on the network's scale, environment, and requirements of network security evaluation to ensure the scientific nature of the indicator selection. Scientific principles make the indicator selection more systematic and objective. Indicators selected based on scientific principles can provide comprehensive, objective, and accurate results for network security situational assessment, facilitating a comprehensive analysis and evaluation of the network security status. Considering the actual situation means that the chosen indicators should be feasible and applicable, adapting to the specific characteristics of the network environment and the evaluation objectives. This ensures that the selected indicators are practical and feasible for practical network security situational assessment.

By referring to the existing indicator system and combined with the existing index selection methods (*Feng-zhu et al., 2015*; *Cai et al., 2022*), the network security situation can be divided into four first-class indexes from different angles, namely, the vulnerability index, threat index, reliability index, and availability index. Among these, the reliability and availability indexes can be subdivided into multiple second-class indexes. The reliability index is divided into network topology score second-class indexes and the number of accessible ports of equipment second-class indexes. The availability indicator can be subdivided into the secondary indicator of traffic change rate, the secondary

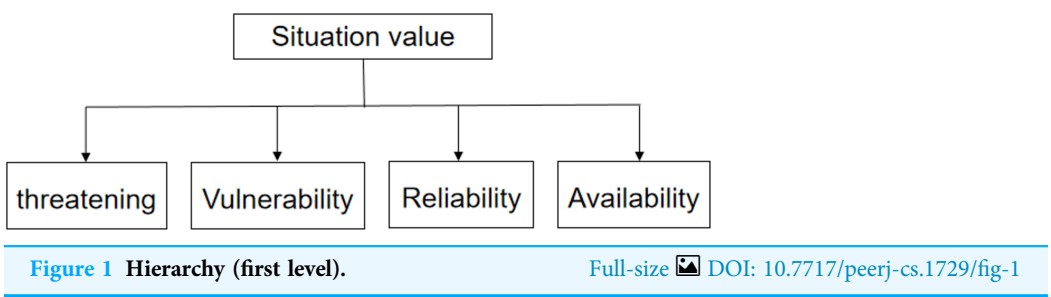

**Figure 1  Hierarchy (first level).**

indicator of data packet distribution, and the secondary indicator of security event frequency.

## Calculation method of situation value based on AHP

After the index system is built, the situation value needs to be calculated. In this regard, the analytic hierarchy process (AHP) may be employed to evaluate network security.

AHP is a quantitative analysis method for decision-making problems under multiple conditions. Moreover, AHP is simple and flexible in application. The main feature of AHP is to establish a hierarchical structure. It can transform human judgment into comparing the importance of several factors. Qualitative judgments that are difficult to quantify can be converted into comparisons of operational importance. In many cases, decision-makers can directly use AHP to make decisions. This greatly improves the effectiveness, reliability, and feasibility of decision-making. The core of AHP is to create a hierarchy of the data for influencing factors. It deconstructs an abstract phenomenon or problem, making it easy to intuitively address complex problems.

AHP has the advantages of simplifying complex problems, simplifying calculation, and reducing subjectivity in the evaluation process, and is widely used. In network security, the analytic hierarchy process is often used to determine the index weight (*Wang et al., 2018*). Here, the AHP calculation process is demonstrated by calculating the situation value using AHP. The process is as follows:

### Establish a hierarchical structure model

A hierarchical structure model was set according to the four indicators (threat, vulnerability, reliability, and availability) established here. The structure chart is shown in Fig. 1.

### Construct judgment matrix

Using the hierarchical structure, the relationship judgment matrix was constructed by comparing the importance of each indicator for the situation value. To put it simply, the indicators were judged in pairs. The Santy's 1–9 scale method is usually used, and the scale method is shown in Table 1.

The relationship judgment matrix constructed is as follows:

$$A = \left( a_{ij} \right)_{4 \times 4}, \quad a_{ij} > 0, \quad a_{ij} = \frac{1}{a_{ji}}, \quad a_{ii} = 1 \tag{1}$$

| Table 1 Meaning of 1–9. | |
|---|---|
| Intensity of importance | Meaning |
| 1 | Indicates that two elements are of equal importance compared |
| 3 | Indicates that the former is slightly more important than the latter |
| 5 | Indicates that the former is obviously more important than the latter |
| 7 | Indicates that the former is extremely important than the latter |
| 9 | Indicates that the former is strongly important than the latter |
| 2, 4, 6, 8 | Indicates the intermediate value of the above adjacent judgments |
| Reciprocal of 1 to 9 | Indicates the importance of the comparison after the exchange order of the corresponding two elements |

## Hierarchical sorting

The hierarchical ranking was used to solve the weight of each indicator according to the judgment matrix formed above.

The power root method was used to calculate the weight:

1) Calculate the 1/m power of the product of each line to obtain an m-dimensional vector, M = 4 in this example, and the formula is as follows:

$$\omega_i' = \sqrt[m]{\prod_{j=1}^{m} a_{ij}} \qquad (2)$$

2) Normalize the vector to obtain the weight vector, and the formula is as follows:

$$\omega_i = \frac{\omega_i'}{\sum_{i=1}^{m} \omega_i'} \qquad (3)$$

## Calculate the maximum feature root and CI value

After the weight matrix was obtained, the maximum eigenvalue was calculated as follows:

$$\lambda_{\max} = \frac{1}{n} \sum_{i=1}^{n} \frac{(AW)_i}{W_i} \qquad (4)$$

where, $n$ is the dimension number. In this example, the dimension number should be 5.

After calculating the maximum eigenvalue, we calculated the $C.I.$ value. The formula is as follows:

$$C.I. = \frac{\lambda_{max}}{n-1} \qquad (5)$$

## The consistency test

The significance of consistency test is to determine whether the constructed relational judgment matrix has logical problems. $R.I.$ values can be derived from Table 2.

**Table 2 R.I. value corresponding to matrix order.**

| 1 | 2 | 3 | 4 | 5 | 6 | 7 | 8 | 9 |
|---|---|---|---|---|---|---|---|---|
| 0 | 0 | 0.58 | 0.90 | 1.12 | 1.24 | 1.32 | 1.41 | 1.45 |

The formula for the *C.R.* value is as follows.

$$C.R. = \frac{C.I.}{R.I.} \tag{6}$$

After calculating the *C.R.* value, if $C.R. < 0.1$, it means that the relationship judgment matrix *A* has passed the consistency test. The calculated weight vector may then be applied.

Using this experiment as an example, the relevant factors first need to be identified, which in this experiment refers to determining the appropriate indicators. Then, a hierarchical structure model must be established. The hierarchical structure of this experiment is shown in Fig. 1. The judgment matrix was constructed and is shown in Eq. (7). The maximum eigenvalue of R was calculated as 4.060433616146346, the CI value was 0.020144538715448707, and the CR value was 0.022634313163425512. Because the CR value was less than 0.1, the judgment matrix passed the consistency test. Finally, the hierarchical ranking and consistency tests were conducted. Calculated $W = \{\text{``}W_T\text{''}, \text{``}W_V\text{''}, \text{``}W_R\text{''}, \text{``}W_U\text{''}\}$. The situation value was calculated using the calculated weight value, and the formula is shown in Eq. (7).

$$R = \begin{bmatrix} 1 & 2 & 2 & 2 \\ 1/2 & 1 & 2 & 2 \\ 1/2 & 1/2 & 1 & 1 \\ 1/2 & 1/2 & 1 & 1 \end{bmatrix}$$

$$SV = W_T T + W_V V + W_R R + W_U U \tag{7}$$

### Indicator calculation process

By referring to the CVSS scoring standard and the established indicator system, as well as the indicator measurement method in this field, and after combining network management experience, the following descriptions and calculations are given in this article.

1) Threat: The threat level refers to various external factors that pose a threat to the network system itself. The most significant impact on network security is typically from external threats, leading to service interruptions or data loss. External threats usually encompass various forms of network attack activities and triggered network security incidents. Therefore, in this study, the main indicators selected under the primary

indicator of threat level are the severity level of attack types, the number of alerts, and the likelihood of security incidents occurring. The formula is as follows:

$$T = \sum_{i=1}^{n} \sum_{j=1}^{k} \frac{10^{D_{ij}} Q_i C_{ij}}{M} \tag{8}$$

where $T$ represents the threat score, $n$ is the total number of hosts in the network system, $k$ is the number of attack types, $M$ is the number of all attacks detected in a period, and $D_{ij}$ is the attack when the i-th host receives the j-th attack the hazard level corresponding to the type, $Q_i$ is the importance of the i-th host, and $C_{ij}$ is the number of times the i-th host receives the attack j.

2) Vulnerability: Vulnerability mainly focuses on describing the inherent security flaws of the network itself. These flaws are primarily manifested in vulnerabilities existing in the network topology, which often become entry points for network attacks. Vulnerability information provides detailed insights into potential security flaws in the network system, including known vulnerability types, potential attack paths, and possible threat levels. On the other hand, the importance of each host reflects its criticality within the network system, indicating how much impact an attack or exploitation of a vulnerability on a specific host could have on the entire system. By considering both factors comprehensively, we can more accurately assess the vulnerability of the network system and identify which vulnerabilities and hosts pose a greater threat to its security. Therefore, vulnerability information and host importance jointly and accurately reflect the network system's vulnerability and serve as secondary indicators under the primary vulnerability indicator. The formula for this indicator is as follows:

$$V = \sum_{i=1}^{n} \sum_{j=1}^{m} Q_i S_{ij} \tag{9}$$

where $V$ stands for vulnerability score, $n$ is the total number of hosts in the network system, $m$ is the number of vulnerability types, $D_{ij}$ is the hazard level corresponding to the attack type when the i-th host receives the j-th attack, $Q_i$ is the importance of the i-th host, $S_{ij}$ and is the score of vulnerability j on the i-th host. This value refers to the CVSS vulnerability score.

3) Reliability: Factors affecting reliability in the network primarily revolve around potential issues with the network system architecture and network devices. Firstly, a well-designed network topology can reduce the probability of network interruptions and enhance network reliability. Additionally, the network topology also influences load balancing and capacity planning, ensuring a reasonable allocation of network resources, thus improving the network system's reliability. Secondly, the number of open ports on devices also has an impact on network system reliability. Open ports serve as communication channels for network devices with the external world, but they also pose potential security vulnerabilities. If too many ports are left open without proper security configuration and management, the network system becomes more vulnerable to

malicious attacks and unauthorized access. This could lead to data breaches, service interruptions, or system crashes, thereby lowering the reliability of the network system. Therefore, the number of open device ports is closely related to network system reliability. In conclusion, a well-structured network topology and an appropriately managed number of open ports on devices can enhance the reliability of the network system. In this study, the network topology score and the number of open ports on devices are primarily used to reflect the status of network reliability. The formula is as follows:

$$R = W_{TP}TP + W_{PR}PR \tag{10}$$

$W_{TP}$ and $W_{PR}$ are the weights of evaluation factors obtained by AHP.

Network topology score: This index reflects the stability of the topology. The higher the index score, the worse the stability of the network system. The formula is as follows:

$$tp_i = \begin{cases} 1.0, & 0 \le br < 3 \\ 0.5, & 3 \le br \le 5 \\ 0.1, & br > 5 \end{cases}$$

$$TP = \sum_{i=1}^{n} tp_i \tag{11}$$

where $TP$ represents the network topology score, $br$ represents the number of branch nodes corresponding to each node in the network system. $tp_i$ represents the topology score corresponding to the i-th node. Generally speaking, the more branches, the more vulnerable to attack, the lower the topology score, and $n$ represents the number of nodes.

The number of accessible device ports: This indicator reflects the number of open ports of all devices in the network. The higher the indicator score, the more open ports, and the worse the network's reliability. The indicator formula is as follows:

$$PR = \sum_{i=1}^{n} pr_i \tag{12}$$

$PR$ represents the number of open ports on the device, $pr_i$ represents the number of open ports on the i-th host, and $n$ represents the number of hosts in the network.

4) Availability: The availability of a network system refers to its ability to function properly and provide the required services during a specific period. In this study, the availability of the current network system was primarily reflected through three indicators: traffic fluctuation rate, data packet distribution, and the frequency of security incidents. Firstly, the traffic fluctuation rate represents the speed of changes in data traffic within the network. A low and stable traffic fluctuation rate indicates that the network load is relatively light and network resources are adequately utilized, thereby improving the network system's availability. However, if the traffic fluctuation rate is significantly high or fluctuates greatly, it may lead to network congestion, increased latency, and a decline in service quality, affecting the network system's availability. Secondly, while the methods of network attacks are complex, any form of network attack requires data packets for

execution. Therefore, the variation in data packet distribution in the network can reveal whether the network is under attack or experiencing unusual conditions. A sudden increase in traffic from abnormal sources within a specific time period may indicate a potential network attack. Finally, the frequency of security incidents refers to how often security events occur within the network. Security incidents may result in service disruptions, data leaks, or malicious attacks, thereby reducing the network's availability. In conclusion, the indicators of traffic fluctuation rate, data packet distribution, and the frequency of security incidents reflect different aspects of the current network system's availability status. The index formula is as follows:

$$U = W_{fr}FR + W_{dp}DP + W_{sr}SR \tag{13}$$

$W_{fr}$, $W_{dp}$, and $W_{sr}$ are the weights of evaluation factors calculated by AHP.

Flow rate of change: The formula is as follows:

$$FR = \frac{f_t}{f_{t-1}} \tag{14}$$

In it, $f_t$ represents the traffic size in the current period, and $f_{t-1}$ represents the traffic size in the previous period.

Packet distribution: The formula is as follows:

$$DP = ip * dtp \tag{15}$$

In the formula, $DP$ represents the distribution of data packets, $ip$ represents the number of different IP addresses, and $dtp$ represents the number of types of different protocols.

The frequency of security incidents: the formula is as follows:

$$SR = \frac{se}{e} \tag{16}$$

$SR$ represents the frequency of security events, $se$ represents the number of security events, and $e$ represents the number of all events.

## Improved whale optimization algorithm based on multiple strategies

The original WOA algorithm had a clear problem: it is easy to fall into a locally optimal solution. Knowing this, we used three improvement methods to resolve this issue.

### WOA

In order to make the evaluation network have a better effect, a large number of network parameters were debugged during the training of the BiGRU network to find the parameter combination with the best effect. However, due to the many BiGRU network parameters, it is not easy to find the optimal parameter combination if the debugging is done manually. Therefore, we designed a parameter optimization method based on MSWOA to find the best parameter combination for BiGRU training. MSWOA is a new
variant of WOA proposed by us. First, introduce the WOA algorithm (*Mirjalili & Lewis, 2016*). The Whale Optimization Algorithm is a novel meta-heuristic optimization algorithm. Meta-heuristic algorithms are improvements on heuristic algorithms, and they are the result of combining random algorithms with local search algorithms. Due to their practicality and ease of use, they have been widely adopted. The calculation steps of WOA are as follows:

The relevant parameters required by the algorithm are set. $t$ is the number of iterations; $p$ is a random number between 0 and 1; 1 is a random number between −1 and 1; $\vec{A}$ and $\vec{C}$ are coefficient vectors; $T$ is the maximum number of iterations; $\vec{r}$ is a random vector in the interval from 0 to 1; $\vec{a}$ is a vector whose value decreases linearly from 2 to 0. The formula for the parameters is as follows:

$$\vec{A} = 2\vec{a} \cdot r - \vec{a}$$
$$\vec{C} = 2 \cdot \vec{r} \tag{17}$$
$$\vec{a} = 2 - \frac{2t}{T}$$

Whales have two strategies to capture their prey: one method is the surrounding the prey, and the other is attacking them with a bubble net. The probability of whales using either method is the same. When $p$ is less than 0.5, the method of surrounding the prey was used; when $p$ is greater than 0.5, the method of attacking with a bubble net was used. The so-called bubble net attack method, in which whales can use spiral bubbles to capture their prey, first calculates the distance between the whale and the prey and then creates a spiral equation through the distance. The mathematical models of these two methods are given below:

$$\vec{D} = \left| \vec{C} \cdot \vec{X}^*(t) - \vec{X}(t) \right| \tag{18}$$

$$\vec{D}' = \left| \vec{X}^*(t) - \vec{X}(t) \right| \tag{19}$$

$$\vec{X}(t+1) = \begin{cases} \vec{X}^*(t) - \vec{A} \cdot \vec{D} & \text{if } p < 0.5 \\ \vec{D}' \cdot e^{bl} \cdot \cos(2\pi l) + \vec{X}^*(t) & \text{if } p \geq 0.5 \end{cases} \tag{20}$$

where $\vec{D}$ and $\vec{D}'$ are the distance between the whale and the prey; $\vec{X}^*(t)$ is the position of the prey; $\vec{X}(t)$ is the whale's position; $b$ is a constant that defines the shape of the spiral.

With the method of encircling the prey, there are two strategies. When $\left| \vec{A} \right|$ is less than 1, the strategy of swimming directly towards the whale in the optimal position is adopted. This strategy refers to Eqs. (18) and (20). When $\left| \vec{A} \right|$ is greater than or equal to 1, the strategy involves swimming towards the position of a random whale, as shown in Eqs. (21) and (22):

$$\vec{D} = \left| \vec{C} \cdot \vec{X}_{rand} - \vec{X} \right| \tag{21}$$

$$\vec{X}(t+1) = \vec{X}_{rand} - \vec{A} \cdot \vec{D} \tag{22}$$

In the above formulae, $\vec{X}_{rand}$ is the position of a randomly selected whale in the swarm.

The WOA algorithm is employed to optimize the trainable parameter vector $\vec{\alpha}$ of the BiGRU model. The input data are the training samples in the normalized evaluation data set. We used Eq. (23) as the fitness function $f_{obj}$ to find the optimal individual in the population, that is, the optimal solution.

$$f_{obj} = \lambda \frac{1}{n} \sum_{i=1}^{n} (y(i) - \hat{y}(i))^2 + \beta \frac{1}{n} \sum_{i=1}^{n} |y(i) - \hat{y}(i)| + \delta \sqrt{\frac{1}{n} \sum_{i=1}^{n} (y(i) - \hat{y}(i))^2} \qquad (23)$$

In Eq. (23), $n$ is the number of training samples; $y$ is the actual value of the network security situation; $\hat{y}$ is the evaluation value of the network security situation calculated according to the current model; $\lambda$, $\beta$ and $\delta$ are coefficients, and the sum of the three is 1, where $\lambda$ is 0.33, $\beta$ is 0.33, and $\delta$ is 0.34.

In the optimization algorithm, $f_{obj}$ is a multi-objective optimization function, and the value of $f_{obj}$ is the comprehensive value of the three performance indicators of the mean square error (MSE), mean absolute error (MAE), and root mean squared error (RMSE). It can avoid the extremes of having an excellent single performance index value with other performance index values that are poor, making the optimization process more reasonable.

The optimization process of WOA is shown in Fig. 2.

## Improvement methods

### Opposition-based learning

Whale populations are typically randomly generated. Random generation means that the population may coalesce around the optimal solution, and the optimization algorithm will find the optimal solution very quickly. However, if the randomly generated population is very far from the optimal solution, it takes a long time to arrive at the optimal solution. The traditional population generation method will easily make the optimization algorithm fall into the optimal local solution if there are multiple locally optimal solutions in the search space. We introduced a method called opposition-based learning (*Mahdavi, Rahnamayan & Deb, 2018*) to prevent the algorithm from falling into an optimal local solution and to improve the optimization efficiency. Opposition-based learning has two primary purposes. One is to speed up the convergence speed of the optimization algorithm. The initial population is generated randomly. Therefore, we can introduce the reverse solution $x_i^*$ to get better results. If the reverse solution of $x_i$ works better than the initial solution, it should be replaced. This way, the reverse solution is 50% more likely to be closer to the optimal global solution than the current solution. Currently, most research on optimization algorithms is applied to reverse learning processes for initializing populations. However, we not only used opposition-based learning in the population initialization process but also applied it to the iterative process of the population. This method also avoids getting stuck in an optimal local solution. When the optimization algorithm reaches a certain stage, it may fall into a locally optimal solution. Replacing the current population with the reverse solution of the population, there is a certain probability of jumping out of the

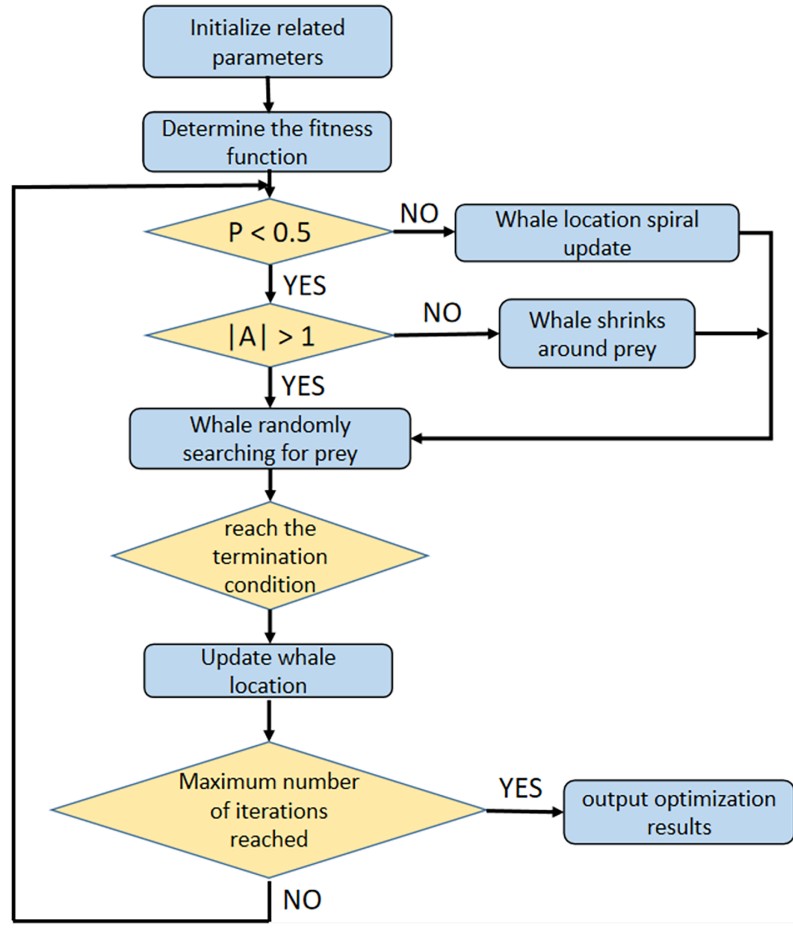

**Figure 2  The optimization process of WOA.**

optimal local solution. Equation (24) is the formula for the multi-dimensional population to find the reverse solution.

$$x_i^* = a_i + b_i - x_i \quad i = 1, \ldots, n \tag{24}$$

Here, $x_i \in [a_i, b_i]$, $a_i$ and $b_i$ are the upper and lower boundaries of the search space, respectively.

After computing the reverse solution, we combine the reverse solution $x_i^*$ with the original solution $x_i$. Assuming that *fitness(x)* is the fitness function in the application scenario, and the number of populations is $N$, combine the reverse solution $x_i^*$ with the initial solution $x_i$ to generate a combination of solution vectors. Calculate the fitness value of all population individuals and then sort the new population according to the size of the fitness value. In our scenario, we prioritize individuals with small fitness values, so we sort them in ascending order and then select the first N individuals with small fitness values as the new population.

### Random adjustment control parameter strategy

In optimization algorithms, striking a good balance between local search and global search capabilities allows the algorithm to maintain a higher search efficiency during the search process (*Cuong-Le et al., 2022*). Simultaneously, it helps to avoid getting trapped in local optima or encountering premature convergence issues, thus enhancing the algorithm's ability to find global optimal solutions. This balancing process typically requires adjusting the algorithm's parameters or designing more sophisticated search strategies to achieve the desired optimization performance.

According to the principle of WOA, the parameter A changes with the change of the convergence factor a, so the value of the convergence factor a plays a vital role in balancing the algorithm's local search and global search capabilities. In the original WOA algorithm, the convergence factor $a$ in Eq. (17) decreases linearly as the number of iterations increases. This variation of the convergence factor tends to make the algorithm fall into a locally optimal solution. If the convergence factor $a$ has more randomness, it will help the algorithm to jump out of the local optimum. This article proposes a strategy for random adjustment of control parameters, and the formula is as follows:

$$a(t) = a_{\min} + \sigma \cdot randn \tag{25}$$

where $a_{min}$ is the minimum value of the convergence factor $a$, $randn$ is a random number that obeys the normal distribution, and the variance $\sigma$ is used to measure the degree of deviation between the convergence factor $a$ and its mathematical expectation (mean value), which can control the error of the convergence factor $a$. Make it evolve in the direction of desired control parameters.

### Levy flight strategy

Levy flight, named after the French mathematician Paul Levy, refers to a random walk in which the probability distribution of the step size is a heavy-tailed distribution (*Yang & Wu, 2009*). There is a relatively high probability of large strides being taken during random walks. Generally speaking, Levy flight is a search method that combines a small step size search and a large step size search. Here, the spiral update strategy of WOA is replaced by the Levy flight strategy. The Levy flight strategy can make the update position more random, jump out of the optimal local solution quickly, and improve the optimization efficiency of the algorithm. The Levy flight strategy has been widely applied to various optimization algorithms (*Minh et al., 2023*). The formula of Levy flight is shown in Eq. (26).

$$x_i^{t+1} = x_i^t + Levy(n) \oplus \alpha \tag{26}$$

Here, $\alpha = x_i^t - x_b$, $\oplus$ is the dot product, $x_i^t$ is the $i$ solution of the $t$ generation, $x_b$ is the current optimal solution, $Levy(s)$ conforms to the Levy distribution, satisfies

$Levy(s) \sim |s|^{-1-\beta}$   $0 < \beta \leq 2$. The mathematical model of Levy distribution is shown in Eq. (27).

$$L(s, \gamma, u) = \begin{cases} \sqrt{\dfrac{\gamma}{2\pi}} exp\left(\left[-\dfrac{\gamma}{2(s-u)}\right]\right) \dfrac{1}{(s-u)^{3/2}} & , \quad 0 < u < s < \infty \\ 0 & , \quad \text{otherwise} \end{cases} \tag{27}$$

In this equation, $\gamma$ and $u$ are greater than 0, $\gamma$ is a proportional parameter, $u$ is a displacement parameter, and the step length of the Levy flight can be expressed as the following formula:

$$Levy(x) = 0.01 \times \frac{u \cdot \sigma}{|v|^{1/\beta}} \tag{28}$$

In it, $u$ and $v$ are standard normal distribution random numbers in [0, 1], $\beta = 1.5$, and the formula of $\sigma$ is as follows:

$$\sigma = \left\{ \frac{\Gamma(1+\beta) \cdot sin\left(\dfrac{\pi\beta}{2}\right)}{\Gamma\left(\dfrac{1+\beta}{2}\right) \cdot \beta \cdot 2^{\left(\frac{\beta-1}{2}\right)}} \right\}^{\frac{1}{\beta}} \tag{29}$$

where $\Gamma(x)$ is the gamma function, namely $\Gamma(x) = (x-1)!$.

## Calculation process of MSWOA

Refer to Algorithm 1 for the calculation process of MSWOA.

## Experiment

### Experimental operating environment and parameter settings

All experiments in this section were conducted using an Intel(R) Xeon(R) PC with CPU E5-2609, 32 GB memory, and 2T hard disk, and the Windows 10 operating system. The operating environment of the program was Python3.9 and pycharm 2021.3.

We used WOA, PSO (*Kennedy & Eberhart, 1995*), GWO (*Mirjalili, Mirjalili & Lewis, 2014*), SSA (*Mirjalili et al., 2017*), and GA (*Whitley, 1994*) algorithms for comparative experiments in order to verify the effect of the improved WOA optimization algorithm. In the experiment, 12 groups of benchmark functions were selected to verify the effect of the improved optimization algorithm. The 12 benchmark functions include six unimodal, five multimodal, and one fixed-dimensional multimodal function. Unimodal functions, multimodal functions and the fixed-dimensional multimodal functions are shown in Table 3. F1–F6 are unimodal functions. The unimodal function has only one global optimal solution and no optimal local solution that is used to test the algorithm's convergence and exploration abilities. F7–F11 are multimodal functions with only one global optimal solution and multiple locally optimal solutions, primarily used to test the

---

**Algorithm 1 MSWOA.**

Input: $fitness(x)$-Fitness function, $N$-population number, $max\_iterations$-maximum iterations, $dim$-dimension of search space, $ub$-upper bound of search space, $lb$-lower bound of search space

Output: Optimum fitness value

1. Initialize whale population $x_i$ $(i = 1, 2, ..., n)$
2. Use opposition-based learning to calculate the reverse solution $x_i^*$ $(i = 1, 2, ..., n)$ of whale population
3. Combine $x_i$ with $x_i^*$ to form a new population $x_i$ $(i = 1, 2, ..., 2n)$
4. Calculate the fitness of each search individual
5. Select the first n search individuals with smaller fitness values, these n individual combinations become a new population $x_i$ $(i = 1, 2, ..., n)$
6. while (t < max_iterations)
7.   for i in $x_i$
8.    Update $\vec{a}, \vec{r}, \vec{A}, \vec{C}, p, l$ /*The value update method of $\vec{a}$ follows the Formula (25) */
9.    if $p < 0.5$
10.     if $|A| < 1$
11.      Update the position of the current individual according to the best individual
12.     else if $|A| \geq 1$
13.      randomly select an individual $x_{rand}$
14.      Update the position of the current individual according to $x_{rand}$
15.     end if
16.    else if $p \geq 0.5$
17.     Update the position of the current individual according to Levy's flight strategy (Formula (26))
18.    end if
19.   end for
20.   Correction of individuals beyond the search scope
21.   Use opposition-based learning to calculate the reverse solution $x_i^*$ $(i = 1, 2, ..., n)$ of whale population
22.   Combine $x_i$ with $x_i^*$ to form a new population $x_i$ $(i = 1, 2, ..., 2n)$
23.   Calculate the fitness of each search individual
24.   Select the first n search individuals with smaller fitness values, these n individual combinations become a new population $x_i$ $(i = 1, 2, ..., n)$
25.   t = t + 1
26. end while
27. return fitness value

---

global search ability of the algorithm. F12 is a fixed-dimensional multipeak test function. The dimension of this function is small, so the optimization pressure is not great. However, the feature of this function is that it is not easy to find the optimal global value, which is mainly used to verify the optimization accuracy of the algorithm. To ensure that there will be no randomness in the experimental results, each optimization algorithm is independently run 30 times on each function. Then the population number of each optimization algorithm is 30, and each optimization algorithm iterates 1,000 times. Finally,

**Table 3  Test function.**

| Function formula | Dim | Range | $F_{min}$ |
|---|---|---|---|
| $F_1(x) = \sum\limits_{i=1}^{30} x_i^2$ | 30 | $[-100, 100]$ | 0 |
| $F_2(x) = \sum\limits_{i=1}^{30}|x_i| + \prod\limits_{i=1}^{30}|x_i|$ | 30 | $[-10, 10]$ | 0 |
| $F_3(x) = \sum\limits_{i=1}^{30}\left(\sum\limits_{j=1}^{i} x_j\right)^2$ | 30 | $[-100, 100]$ | 0 |
| $F_4(x) = max\{|x_i|, 1 \le i \le 30\}$ | 30 | $[-100, 100]$ | 0 |
| $F_5(x) = \sum\limits_{i=1}^{30}(\lfloor x_i + 0.5\rfloor)^2$ | 30 | $[-100, 100]$ | 0 |
| $F_6(x) = \sum\limits_{i=1}^{30} ix_i^4 + random[0, 1)$ | 30 | $[-1.28, 1.28]$ | 0 |
| $F_7(x) = -\sum\limits_{i=1}^{30}\left(x_i \sin\left(\sqrt{|x_i|}\right)\right)$ | 30 | $[-500, 500]$ | $-12{,}569.5$ |
| $F_8(x) = \sum\limits_{i=1}^{30}\left[x_i^2 - 10\cos(2\pi x_i) + 10\right]$ | 30 | $[-5.12, 5.12]$ | 0 |
| $F_9(x) = -20\exp\left(-0.2\sqrt{\dfrac{1}{30}\sum\limits_{i=1}^{30} x_i^2}\right) - \exp\left(\dfrac{1}{30}\sum\limits_{i=1}^{30}\cos 2\pi x_i\right) + 20 + e$ | 30 | $[-32, 32]$ | 0 |
| $F_{10}(x) = \dfrac{1}{4{,}000}\sum\limits_{i=1}^{30} x_i^2 - \prod\limits_{i=1}^{30}\cos\left(\dfrac{x_i}{\sqrt{i}}\right) + 1$ | 30 | $[-600, 600]$ | 0 |
| $F_{11}(x) = 0.1\left\{\begin{array}{l} \sin^2(\pi 3x_1) + \\ \sum\limits_{i=1}^{29}(x_i - 1)^2[1 + \sin^2(3\pi x_{i+1})] + \\ (x_n - 1)^2[1 + \sin^2(2\pi x_{30})] \end{array}\right\} + \sum\limits_{i=1}^{30} u(x_i, 5, 100, 4)$ | 30 | $[-50, 50]$ | 0 |
| $u(x_i, a, k, m) = \begin{cases} k(x_i - a)^m, & x_i > a \\ 0, & -a \le x_i \le a \\ k(-x_i - a)^m, & x_i < -a \end{cases}$ <br> $y_i = 1 + \dfrac{1}{4}(x_i + 1)$ | | | |
| $F_{12}(x) = \sum\limits_{i=1}^{11}\left[a_i - \dfrac{x_1(b_i^2 + b_i x_2)}{b_i^2 + b_i x_3 + x_4}\right]^2$ | 4 | $[-5, 5]$ | 0.0003075 |

the optimal value (Best), the average value (Ave), and the standard deviation (Sd) of each optimization algorithm on the benchmark function are obtained.

### Optimal precision analysis

The optimization results of the improved algorithm proposed here and the optimization results of other classic optimization algorithms are shown in Table 4. The hybrid strategy improved WOA and has apparent advantages in convergence accuracy on the benchmark functions F1, F2, F3, F4, and F5, compared with different classical algorithms. The advantages are high convergence accuracy and high stability. On the benchmark function F6, the convergence accuracy of the MSWOA algorithm was significantly higher than that of GWO, GA, PSO, and SSA. However, the optimal value of the convergence accuracy was

**Table 4  Optimal results.**

| Function | Algorithm | Best | Ave | Sd |
|---|---|---|---|---|
| F1 | GWO | 1.01E−067 | 1.40E−065 | 3.95E−065 |
| | GA | 1.04E+003 | 1.53E+003 | 4.50E+002 |
| | PSO | 1.00E−008 | 2.84E−009 | 6.12E−009 |
| | SSA | 1.01E−008 | 1.31E−008 | 3.37E−009 |
| | WOA | 1.00E−153 | 2.81E−148 | 1.51E−147 |
| | MSWOA | 0.00E+000 | 6.87E−264 | 0.00E+000 |
| F2 | GWO | 1.02E−038 | 1.50E−038 | 1.92E−038 |
| | GA | 1.10E+001 | 1.59E+001 | 3.54E+000 |
| | PSO | 1.50E−004 | 7.67E+000 | 8.03E+000 |
| | SSA | 5.39E−003 | 7.73E−001 | 6.56E−001 |
| | WOA | 1.08E−104 | 7.50E−104 | 2.10E−103 |
| | MSWOA | 1.05E−237 | 1.89E−199 | 0.00E+000 |
| F3 | GWO | 1.01E+04 | 7.05E+03 | 3.35E+03 |
| | GA | 2.72E+04 | 5.08E+04 | 1.75E+04 |
| | PSO | 1.09E+04 | 7.39E+03 | 3.38E+03 |
| | SSA | 1.00E+04 | 1.74E+04 | 1.03E+04 |
| | WOA | 1.25E+06 | 1.24E+06 | 8.01E+05 |
| | MSWOA | 1.09E−28 | 7.38E−13 | 2.77E−12 |
| F4 | GWO | 1.00E−12 | 5.58E−13 | 1.47E−12 |
| | GA | 2.39E+01 | 3.49E+01 | 4.51E+00 |
| | PSO | 3.16E−01 | 6.30E−01 | 1.89E−01 |
| | SSA | 1.03E+01 | 7.29E+00 | 3.69E+00 |
| | WOA | 3.05E−02 | 4.68E+01 | 2.88E+01 |
| | MSWOA | 1.00E−27 | 2.02E−14 | 1.08E−13 |
| F5 | GWO | 0.00E+00 | 0.00E+00 | 0.00E+00 |
| | GA | 6.88E+02 | 1.41E+03 | 5.30E+02 |
| | PSO | 0.00E+00 | 0.00E+00 | 0.00E+00 |
| | SSA | 0.00E+00 | 3.70E+00 | 3.41E+00 |
| | WOA | 0.00E+00 | 3.00E−02 | 1.80E−01 |
| | MSWOA | 0.00E+00 | 0.00E+00 | 0.00E+00 |
| F6 | GWO | 1.60E−04 | 1.09E−03 | 7.50E−04 |
| | GA | 3.71E−01 | 8.45E−01 | 2.34E−01 |
| | PSO | 2.31E−02 | 3.98E+00 | 5.44E+00 |
| | SSA | 3.84E−02 | 8.21E−02 | 3.23E−02 |
| | WOA | 1.20E−04 | 2.59E−03 | 3.43E−03 |
| | MSWOA | 1.11E−04 | 5.70E−04 | 7.80E−04 |
| F7 | GWO | −8.30E+03 | −5.91E+03 | 1.13E+03 |
| | GA | −1.03E+04 | −9.40E+03 | 5.83E+02 |
| | PSO | −7.34E+03 | −6.01E+03 | 1.03E+03 |
| | SSA | −9.06E+03 | −7.64E+03 | 7.16E+02 |
| | WOA | −1.26E+04 | −1.08E+04 | 1.77E+03 |
| | MSWOA | −1.26E+04 | −1.24E+04 | 4.09E+02 |

(Continued)

| Table 4 (continued) | | | | |
|---|---|---|---|---|
| Function | Algorithm | Best | Ave | Sd |
| F8 | GWO | 0.00E+00 | 2.97E+00 | 5.57E+00 |
| | GA | 3.86E+01 | 6.16E+01 | 8.86E+00 |
| | PSO | 3.78E+01 | 9.01E+01 | 29.35E+00 |
| | SSA | 3.18E+01 | 5.90E+01 | 1.66E+01 |
| | WOA | 0.00E+00 | 0.00E+00 | 0.00E+00 |
| | MSWOA | 0.00E+00 | 0.00E+00 | 0.00E+00 |
| F9 | GWO | 1.47E−14 | 1.73E−14 | 3.17E−15 |
| | GA | 1.01E+01 | 1.04E+01 | 1.22E+00 |
| | PSO | 1.01E−04 | 8.17E−05 | 1.56E−04 |
| | SSA | 1.16E+00 | 2.08E+00 | 8.76E−01 |
| | WOA | 4.00E−15 | 4.12E−15 | 2.67E−15 |
| | MSWOA | 4.00E−15 | 7.99E−16 | 1.07E−15 |
| F10 | GWO | 0.00E+00 | 2.07E−03 | 4.22E−03 |
| | GA | 1.01E+00 | 1.03E+00 | 1.28E−02 |
| | PSO | 2.97E−13 | 8.46E−03 | 8.61E−03 |
| | SSA | 1.07E−10 | 1.03E−02 | 1.01E−02 |
| | WOA | 0.00E+00 | 5.16E−03 | 1.55E−02 |
| | MSWOA | 0.00E+00 | 0.00E+00 | 0.00E+00 |
| F11 | GWO | 1.99E−01 | 4.96E−01 | 1.64E−01 |
| | GA | 2.56E+01 | 2.13E+05 | 5.01E+05 |
| | PSO | 2.69E−01 | 8.32E−01 | 3.57E−01 |
| | SSA | 8.50E−10 | 1.07E+00 | 5.53E+00 |
| | WOA | 3.40E−02 | 2.59E−01 | 2.52E−01 |
| | MSWOA | 5.08E−12 | 4.76E−03 | 8.83E−03 |
| F12 | GWO | 3.10E−04 | 2.36E−03 | 6.01E−03 |
| | GA | 6.77E−04 | 1.32E−03 | 5.21E−04 |
| | PSO | 3.43E−04 | 7.45E−03 | 8.64E−03 |
| | SSA | 3.62E−04 | 1.53E−03 | 3.51E−03 |
| | WOA | 3.08E−04 | 7.21E−04 | 4.43E−04 |
| | MSWOA | 3.08E−04 | 6.50E−04 | 2.94E−04 |

close to the optimal value of the WOA algorithm, indicating that the optimization potential of WOA on this function was close to the MSWOA algorithm. However, the average and standard deviation of the convergence precision of MSWOA was an order of magnitude lower than that of WOA, indicating that the convergence stability of MSWOA was higher than that of WOA. On the benchmark functions F10 and F11, the convergence accuracy of MSWOA was significantly higher than other optimization algorithms, and the stability was also significantly higher than other algorithms. In functions F7, F8, and F9, the optimal value of the convergence precision of MSWOA was close to the optimal value

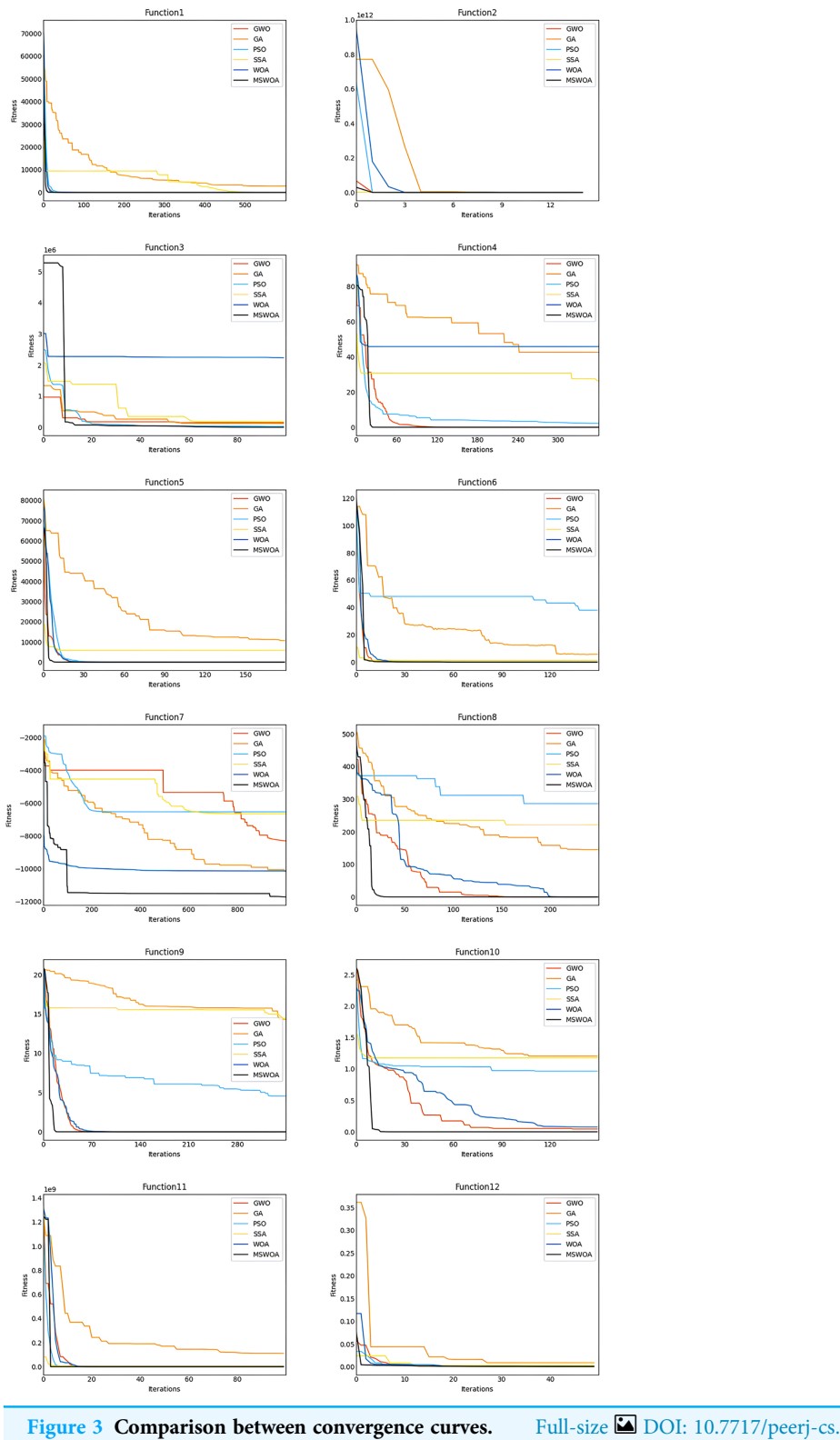

**Figure 3 Comparison between convergence curves.**

of other optimization algorithms, indicating that different algorithms can find the optimal global solution under certain conditions. However, the average and standard deviation of the convergence accuracy of MSWOA are smaller than other algorithms, indicating that MSWOA is more stable than other algorithms. In function F12, the optimal value, the average value, and the standard deviation of the convergence accuracy of MSWOA are better than other algorithms, indicating that the optimization performance of MSWOA is excellent.

### Convergence speed analysis

First, we analyzed the convergence curves of the optimization algorithm for the functions F1, F2, F3, F4, F5, and F6. Figure 3 shows that the convergence speed of MSWOA in the F1 function was faster than that of GA and SSA, and the convergence speed of MSWOA was slightly faster than that of PSO, WOA, and GWO. On the F2 function, the convergence speed of MSWOA was faster than that of GA, WOA, and PSO but slightly slower than that of SSA. In functions F3 to F5, the convergence speed of MSWOA was faster than that of the other five optimization algorithms. In the F6 function, the convergence speed of MSWOA was faster than that of PSO, GA, and WOA and was close to that of SSA and GWO. The functions F1 to F6 are all unimodal functions. The unimodal function has only one global optimal value and no optimal local value. This shows that the algorithm's performance on the unimodal function can reflect the convergence ability of the algorithm. In general, the convergence speed of MSWOA was significantly faster than other optimization algorithms for the unimodal function, indicating that the convergence performance of MSWOA is excellent. Then analyze the convergence curves of the optimization algorithm on the F7, F8, F9, F10, F11, and F12 functions. Figure 3 shows that on functions F7 to F12, the convergence speed of MSWOA was significantly faster than that of the other algorithms. F7 to F12 are multimodal functions. The multimodal function has an optimal global value and multiple local optimal values. The algorithm's performance on the multimodal function may reflect the algorithm's global search and local optimization capabilities. The analysis shows that MSWOA performed better than other global search and local optimization algorithms, which indicated that the ability of MSWOA's to solve complex optimization problems was excellent.

## THE PROPOSED NSSA MODEL AND ALGORITHM

### BiGRU

BiGRU and GRU essentially work on the same principle. GRU is introduced first.

The gate recurrent unit (GRU) is a variant of the LSTM network. Moreover, the GRU network is more straightforward in structure than the LSTM network. GRU network controls information updating through gating method. Unlike LSTM, GRU does not introduce additional memory cells. The update gate is an important part of the GRU network. The update gate can control how much information the current evaluation status obtains from the historical evaluation status and how much updated information the

current evaluation status needs to receive from the candidate evaluation status. The above description is available in Eq. (30):

$$h_t = z_t \odot h_{t-1} + (1 - z_t) \odot g(x_t, h_{t-1}; \theta) \qquad (30)$$

$$z_t = \sigma(W_z x_t + U_z h_{t-1} + b_z) \qquad (31)$$

where $x_t$ is the data set of network security situation assessment; $z_t$ is the update gate (refer to Eq. (31)); $h_t$ is the current evaluation status; $h_{t-1}$ is the last evaluation state, and $W_z$, $U_Z$, and $b_z$ in Eq. (31) are learnable parameters. There is a certain amount of redundancy and complementarity between input gates and forgetting gates in the LSTM network. GRU networks directly use gates to control the balance between input and forgetting. When $Z_t = 0$, there is a nonlinear functional relationship between the current assess state $h_t$ and the previous assess state $h_{t-1}$; When $z_t = 1$, the relationship between $h_t$ and $h_{t-1}$ is linear. The function $g(x_t, h_{t-1}; \theta)$ in Eq. (32) is defined as follows.

$$g(x_t, h_{t-1}; \theta) = \tilde{h}_t = \tan h(W_h x_t + U_h(r_t \odot h_{t-1}) + b_h) \qquad (32)$$

where $\tilde{h}_t$ represents the candidate evaluation state at the current moment, and $r_t$ is the reset gate, which controls whether the calculation of the candidate evaluation state $\tilde{h}_t$ depends on the evaluation state $h_{t-1}$ at the last moment. The formula of the reset gate is as follows:

$$r_t = \sigma(W_r x_t + U_r h_{t-1} + b_r) \qquad (33)$$

When $r_t = 0$, the candidate evaluation state $\tilde{h}_t$ is only related to the current input $x_t$ and has nothing to do with the historical evaluation state $h_{t-1}$. When $r_t = 1$, the candidate evaluation state $\tilde{h}_t$ is related to the current input $x_t$ and the historical evaluation state $h_{t-1}$, which is consistent with the simple recurrent network.

In summary, the state update method of the GRU network is as follows:

$$h_t = z_t \odot h_{t-1} + (1 - z_t) \odot \tilde{h}_t \qquad (34)$$

It can be seen that when $z_t = 0$, $r = 1$, the GRU network degenerates into a simple recurrent network; when $z_t = 0$, $r = 0$, the current state $h_t$ is only related to the current input $x_t$ and has nothing to do with the historical state $h_{t-1}$. When $z_t = 1$, the current state $h_t$ is equal to the previous state $h_{t-1}$, independent of the current input $x_t$.

The recurrent unit structure of the GRU network is shown in Fig. 4.

In some tasks, a moment's output is related to the information of the past moment and the information of the next moment. For example, in a given a sentence, the part of speech of a word is determined by its context. That is, it contains information on the left and right sides. In these tasks, the network's capabilities are enhanced by including a network layer that transmits information in reverse chronological order.

The bidirectional gated recurrent unit (BiGRU) consists of two layers of GRUs with the same input but different directions of information transfer.

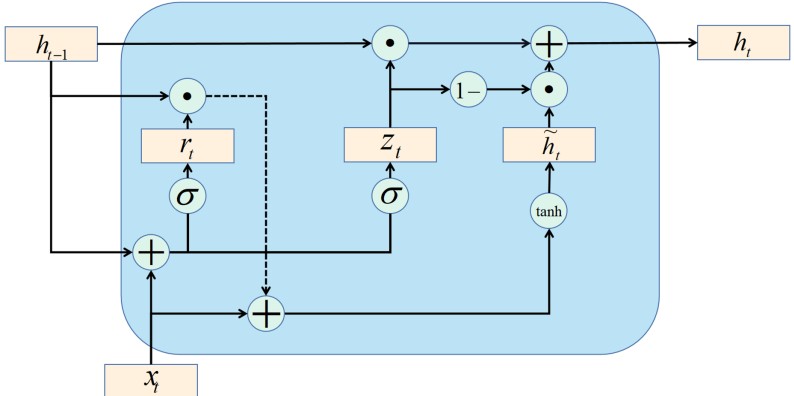

**Figure 4 GRU structure.**

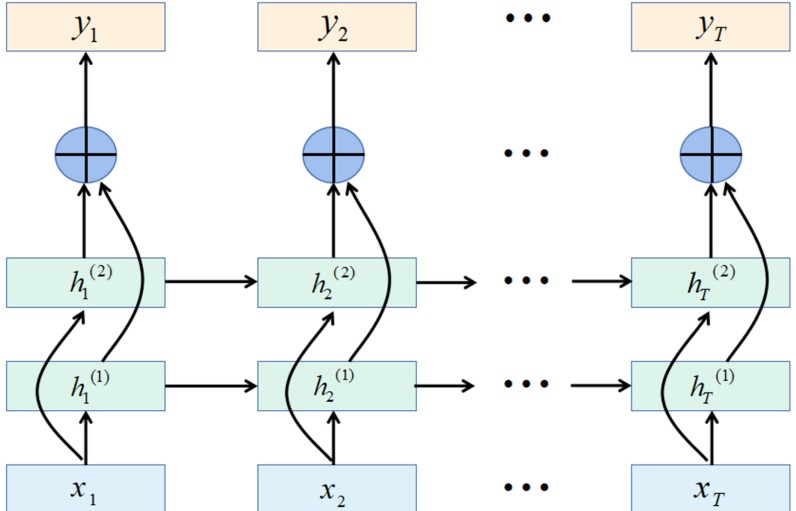

**Figure 5 Bidirectional recurrent neural network structure.**

Assuming that the first layer is in time order, the second layer is in reverse time order, the hidden states at time t are defined as $h_t^{(1)}$ and $h_t^{(2)}$, Eqs. (35) and (36) are the definitions of the two hidden states, Eq. (37) is the definition for the current state.

$$h_t^{(1)} = f\left(U^{(1)}h_{t-1}^{(1)} + W^{(1)}x_t + b^{(1)}\right) \tag{35}$$

$$h_t^{(2)} = f\left(U^{(2)}h_{t+1}^{(2)} + W^{(2)}x_t + b^{(2)}\right) \tag{36}$$

$$h_t = h_t^{(1)} \oplus h_t^{(2)} \tag{37}$$

where $\oplus$ is the vector concatenation operation.

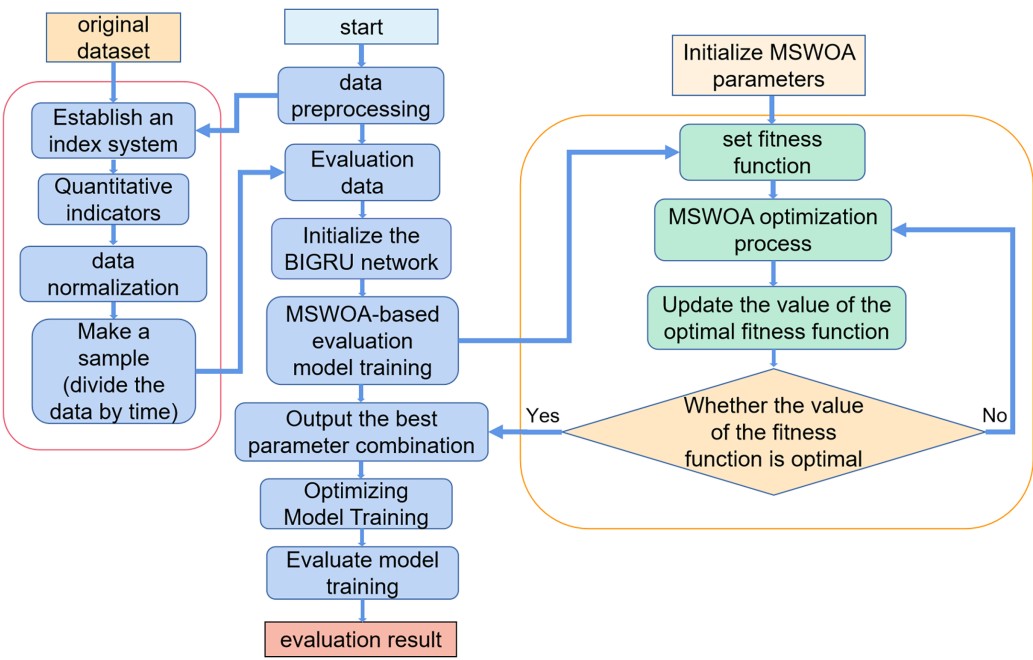

**Figure 6** **MSWOA-BiGRU network security situation assessment method process.**

The bidirectional recurrent neural network expanded by time is shown in Fig. 5.

## The proposed NSSA model and algorithm

The MSWOA-BiGRU model proposed for NSSA is divided into the following three steps.
1. Establish a situational assessment index system.

We divided the network security situation into four first-level indicators: vulnerability, threat, reliability, and availability. Reliability is determined by two second-level indicators, namely the network topology score and the number of open ports on the device, while availability is determined by three secondary indicators: the rate of change in traffic, the distribution of data packets, and the frequency of security incidents.
2. Make samples.

All the index values are calculated from the original data according to the quantization formula, then normalized, and made into a corresponding time slice library. Part of the time slice is extracted from the time slice library as training samples, and the remaining time slices are used as test samples.
3. Training model.

The training samples are input into the MSWOA-BiGRU model for training, which is divided into two parts: optimization and evaluation. Use the BiGRU network to evaluate the network security situation and get the situation evaluation value. MSWOA is used to optimize the evaluation process and improve the evaluation effect.

**Algorithm 2** Network security situation assessment algorithm based on the MSWOA-BiGRU model.

Input: $D$-data

Output: Evaluation value, loss_value, optimal parameter combination

1. get $D_{train}$, $D_{test}$ from $D$
2. Call optimization function MSWOA
3. Set $max\_iterations$, $noposs$, $noclus$
4. Initialize $poss\_sols1$, $poss\_sols2$
5. Initialize $gbest1$, $gbest2$
6. for it in 0:$max\_iterations$ do
7.     for i in 0:$noposs$ do
8.         Update $\vec{a}$, $\vec{r}$, $\vec{A}$, $\vec{C}$, $p$, $l$
9.         Update whale position (parameter vector) /*Formular (20), (22) and (26)*/
10.         Call fitness function, and input whale position into fitness function
11.         Call the built $BIGRU$ network
12.         The components in the hyperparameter ←parameter vector of BiGRU network
13.         Training $BIGRU$ Network
14.         Return Value of fitness function ←fitness function calculation formula /* Formula (23)*/
15.         Compare the value of the current fitness function $fitnessi$ with the value of the current optimal $global\_fitness$
16.         if $fitnessi < global\_fitness$ do
17.             $global_{fitness} \leftarrow fitnessi$
18.             Predator position $gbest1$, $gbest2$ ←current whale position $poss\_sols1$, $poss\_sols2$
19.         end if
20.     end for
21. end for
22. Return optimal parameter combination, loss_value

The flow of the NSSA method based on MSWOA optimization BiGRU proposed is shown in Fig. 6. The specific algorithm (Algorithm 2) is as follows:

# DATASETS AND EVALUATION CRITERIA

## Experimental environment and datasets

The experiments all used Python 3.9 and Pytorch 1.9.1 to build the neural network model, used Pycharm 2021.3 to compile the model, and used the CPU to train the model. The experiments were performed on a PC with Intel(R) Xeon(R) CPU E5-2609, 32 GB RAM, Windows 10.

This article selects the UNSW-NB15 dataset, which is relatively authoritative in the field of network security research, as the source dataset for evaluation (the UNSW-NB15 dataset can be downloaded from https://research.unsw.edu.au/projects/unsw-nb15-dataset). This dataset can be viewed as a continuous time series. The dataset was a continuous time series.

**Table 5 UNSW-NB15 dataset information.**

| Types of aggressive behavior | Training set | Test set |
|---|---|---|
| Normal | 56,000 | 37,000 |
| Exploits | 33,393 | 11,132 |
| Fuzzers | 18,184 | 6,062 |
| Generic | 40,000 | 18,871 |
| DoS | 12,264 | 4,089 |
| Reconnaissance | 10,491 | 3,496 |
| Analysis | 2,000 | 677 |
| Backdoor | 1,746 | 583 |
| Shellcode | 1,133 | 378 |

**Table 6 Index system of NSSA.**

| Indicator classification | Indicator name |
|---|---|
| Vulnerability | Vulnerability |
| Threatening | Threatening |
| Reliability | Network topology score |
| | Number of open ports on the device |
| Availability | Flow rate of change |
| | Packet distribution |
| | Frequency of security incidents |

It was divided into two periods of time. The first interval was recorded as 2015-1-22 19:50 to 2015-1-23 8:25, and the second was 2015-2-18 8:23 to 2015-2-18 20:21. This dataset has several advantages. Firstly, the dataset has nine attack types, which are more in line with the actual network environment. Secondly, the dataset has vulnerability information. Finally, compared with the traditional dataset, this dataset has more diverse attack characteristics. This dataset has 49 features (*Moustafa & Slay, 2015*). The details of the dataset are shown in Table 5.

To more accurately assess the network security situation, we processed the dataset. First, based on the existing time information of the UNSW-NB15 dataset, we divided it into multiple time slices at 1-min intervals. Then, we used the metric extraction and quantification methods mentioned earlier to process each time slice. After processing, we obtained the situation assessment metrics, as shown in Table 6. After constructing the metrics, we used the situation value calculation methods mentioned in the previous chapters to calculate the actual situation values for each time slice.

This study refers to the basic cybersecurity situation index of the Multi-State Information Sharing and Analysis Center (MS-ISAC) and the National Overall Contingency Plan for Public Emergencies. It divides the network security situations into

**Table 7 Grade and meaning of network security situation assessment value.**

| Situation level | Situation value range | Level meaning |
|---|---|---|
| Safety | [0, 0.2) | The network environment is stable and routine, and no security loopholes have been found. |
| Low risk | [0.2, 0.4) | The network environment was slightly affected, illegal activities such as network attacks occurred, and low-risk security vulnerabilities were discovered. |
| Medium risk | [0.4, 0.6) | The network environment has been affected to a certain extent, the number of network attacks has increased, and some medium-risk vulnerabilities have been discovered. |
| High risk | [0.6, 0.8) | The network environment has been greatly affected, the types and numbers of other illegal activities such as network attacks have increased, and high-risk vulnerabilities have been discovered. |
| Serious | [0.8, 1.0) | The network environment has been seriously affected, with a large number of attacks and high-risk vulnerabilities appearing. |

five levels: safe, low-risk, medium-risk, high-risk, and serious. The specific information shown in Table 7.

## Experimental evaluation criteria

Connecting the network security situational assessment values and the corresponding true values for each time period can generate the network security situational assessment value curve and the true value curve. The degree of fit between these two curves essentially reflects the accuracy of the evaluation. The purpose of selecting the following four metrics is to measure the degree of fit, essentially evaluating the effectiveness of the model. The performance metrics are as follows:

Mean square error (MSE), which is the squared difference between the actual and predicted values, is calculated as follows:

$$MSE = \frac{1}{m}\sum_{i=1}^{m}(y_i - \hat{y}_i)^2 \tag{38}$$

Mean absolute error (MAE), which represents the mean value of the absolute error between the predicted value and the observed value, is calculated as follows:

$$MAE = \frac{1}{m}\sum_{i=1}^{m}\left|(y_i - \hat{y}_i)\right| \tag{39}$$

Root mean square error (RMSE), which represents the sample standard deviation of the difference between predicted and observed values (residuals), is calculated as follows:

$$RMSE = \sqrt{\frac{1}{m}\sum_{i=1}^{m}(y_i - \hat{y}_i)^2} \tag{40}$$

R$^2$ (R$^2$ score) represents the proportion of all the dependent variable variations that the independent variable can explain through the regression relationship. The formula is as follows:

$$R^2 = 1 - \frac{\sum_{i=1}^{m} \left(y_i - \hat{y}_i\right)^2}{\sum_{i=1}^{m} \left(y_i - \bar{y}\right)^2} \qquad (41)$$

The smaller the values of MSE, MAE, and RMSE, the more accurate the results of the model, and the larger the value of the R$^2$ index, the more precise the results of the model.

## EXPERIMENT AND ANALYSIS

### Parameters

By default, in MSWOA-BiGRU, the search space for the learning rate is between 0.0001 and 0.01, the search space for batch size is between 8 and 64, the search space for epochs is between 150 and 200, the search space for the number of neurons in the GRU hidden layer is between 5 and 20, and the number of neurons in the fully connected layer is between 5 and 100. The maximum number of iterations for the improved optimization algorithm MSWOA is 200 rounds, with a population size of 30. The default loss function is mse loss, and the optimizer is adam.

### Index system construction experiment

The index value is calculated according to the index system and index calculation method constructed in this article. The index value was normalized to eliminate the dimensional influence between the indexes. Finally, the network security situation value was calculated using AHP. We selected some samples from the test samples to verify the effectiveness of the indicator system construction method.

The number of high-risk vulnerabilities in the network system may represent the network security situation, to a certain extent. The larger the number of high-risk vulnerabilities are, the more serious the security risk of the net system are. Figure 7 shows the situation value trend calculated according to the index system proposed in this article. Figure 8 shows trends in the number of high-risk, medium-risk, and low-risk vulnerabilities.

By comparing Figs. 7 and 8, it can be seen that the extreme value of the number of high-level vulnerabilities and the extreme value of the situation value are the same as the sample points. For example, at sample point 1, the number of high-level vulnerabilities in Fig. 8 is a local maximum value, and the situation value in Fig. 7 is a local maximum. In Fig. 8, sample points 6, 13, and 17 are all extreme points. The situation value of the same sample point in Fig. 7 is also an extreme point. It is similar to the trend in the number of high-level vulnerabilities and the trend in the value of the situation. For example, from sample point 6 to 11 and from sample point 15 to 19, the change trends in Figs. 7 and 8 are the same during these two periods. The above examples show that our index system is effective.

Figure 9 shows the trend of changes in the number of security events. A similar trend can be seen in Figs. 7 and 8. However, some periods cannot be perfectly matched since

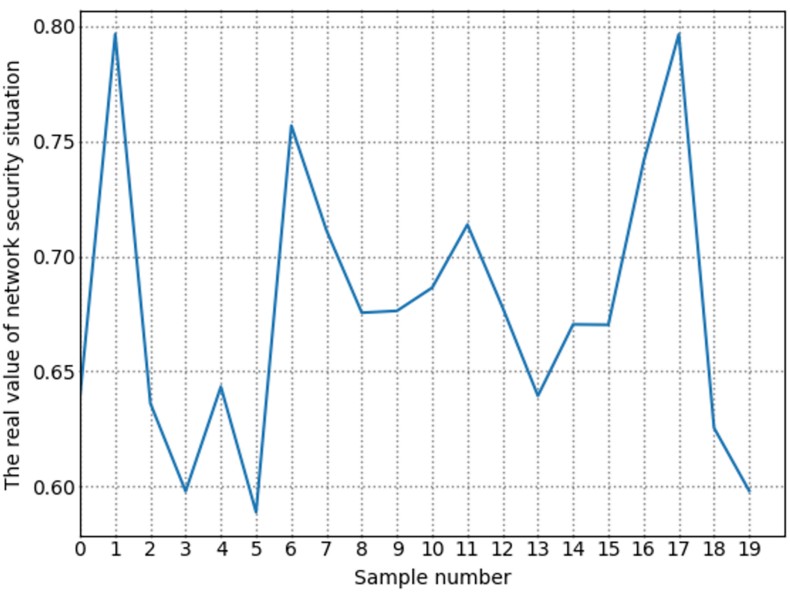

**Figure 7  Change trend of true value of network security situation.**

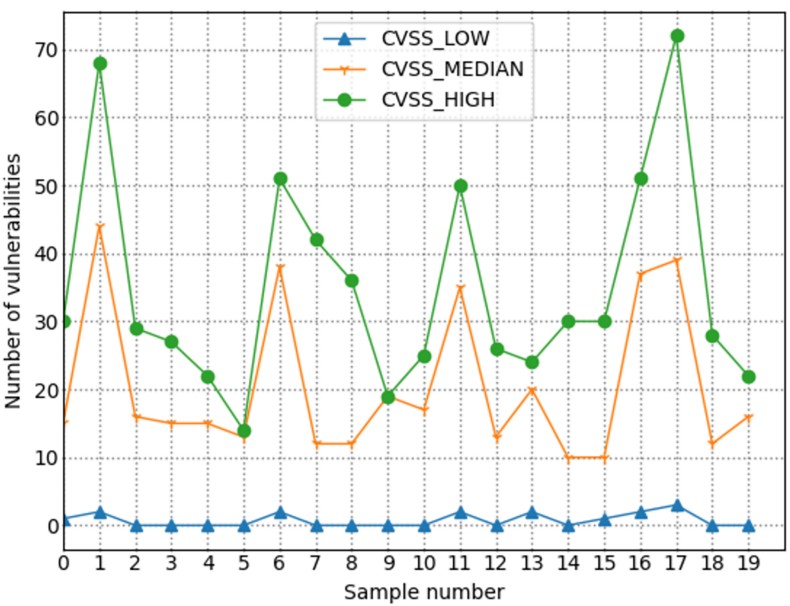

**Figure 8  Trends in the number of vulnerabilities with different severity levels.**

many factors can determine the size of the situation value. For example, from sample point 3 to sample point 5, during this period, the overall trend of change in the situation value decreases. Still, from sample point 3 to sample point 4, the trend increases, but as shown in Fig. 8, the sample number of high-risk vulnerabilities decreased from point 3 to point

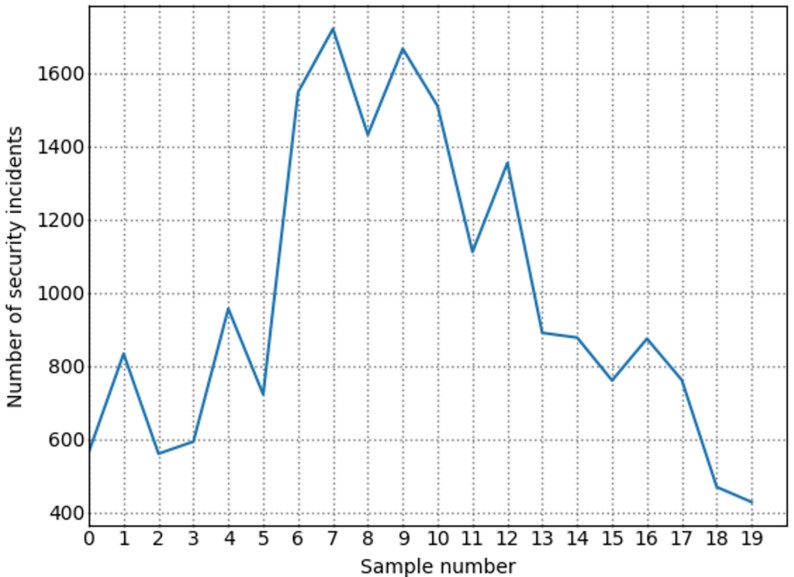

**Figure 9 Trend of the number of security incidents.**

4. Referring to Fig. 9, we can find that the number of attacks from sample point 3 to sample point 4 increased, and the number of security incidents suddenly rose. As a result, the network system faces more severe risks, which results in a higher situation value. For example, during the period from sample point 8 to sample point 10, on the whole, the situation value during this period increased. However, Fig. 8 shows that the number of high-risk vulnerabilities has decreased during this period. The number of security incidents increased during this period (Fig. 9), therefore, it is reasonable that the situation value will be higher. By comparing Figs. 7–9, we further verified the rationality of the index system constructed in this study. The situation value calculated according to the index construction system proposed in this study was found to be in line with the real-life application.

## Analysis of network security situation assessment results

Through four experiments, the experimental results and analytical results proved the effectiveness of the designed BiGRU model based on MSWOA optimization. The experimental data are the average results of the model running ten times. Experiment I aims to verify the excellent performance of BiGRU by comparing the classical machine learning algorithms. The goal of experiment II was to prove that the version of the model after adding the optimization algorithm is better than that of the model without the optimization algorithm. Experiment III's aim was to verify that the performance of MSWOA-BiGRU optimization proposed was better than other algorithms. To verify that the model proposed can evaluate the overall condition of network security more effectively, experiment IV compared the changing trends of the situation values across the eight models introduced here.

**Table 8 Comparison of evaluation effects of GRU, LSTM and BiGRU.**

| MODEL | MSE | MAE | RMSE | $R^2$ |
|---|---|---|---|---|
| GRU | 0.003592 | 0.051028 | 0.059930 | 0.768734 |
| LSTM | 0.003236 | 0.047526 | 0.05688 | 0.791650 |
| BiGRU | 0.002530 | 0.035317 | 0.05030 | 0.837116 |

**Table 9 Improvement of BiGRU.**

| Model | Object to be compared | MSE improvement | MAE improvement | RMSE improvement | $R^2$ improvement |
|---|---|---|---|---|---|
| BiGRU | GRU | 29.57% | 30.79% | 16.07% | 8.90% |
| BiGRU | LSTM | 21.82% | 25.69% | 11.57% | 5.74% |

**Table 10 Comparison of the evaluation effect between the optimized evaluation model and BiGRU.**

| MODEL | MSE | MAE | RMSE | $R^2$ |
|---|---|---|---|---|
| WOA+GRU | 0.002154 | 0.034556 | 0.046415 | 0.861281 |
| PSO+GRU | 0.002038 | 0.034123 | 0.045140 | 0.868797 |
| WOA+BiGRU | 0.001718 | 0.032079 | 0.041454 | 0.889346 |
| PSO+BiGRU | 0.002045 | 0.033526 | 0.045222 | 0.868316 |
| BiGRU | 0.002530 | 0.035317 | 0.05030 | 0.837116 |
| **MSWOA+BIGRU** | **0.001392** | **0.028871** | **0.037309** | **0.910370** |

Note:
Results for the proposed model are shown in bold.

### Experiment I: comparison of classical machine learning algorithms in evaluation effect

The model proposed in this study is BiGRU model optimized by MSWOA. In order to show that the evaluation effect of the model was excellent, the performance of the basic model BiGRU needed to be tested first. Compared with the GRU model, the average MSE, MAE, RMSE, and $R^2$ of the BiGRU model increased by 29.57%, 30.79%, 16.07%, and 8.90%, respectively (Table 8). Compared with the LSTM model, the average MSE and MAE of the regular BiGRU model increased by 21.82%, 25.69%, RMSE increased by 11.57%, and $R^2$ increased by 5.74%, respectively. Hence, the evaluation effect of BiGRU was superior to other classical machine learning algorithms. The visual representation of the improvement degree is shown in Table 9.

### Experiment II: comparison of evaluation effects of deep learning algorithms based on swarm intelligence optimization algorithms

To test whether the swarm intelligent optimization algorithms are effective for deep learning, experiment II is designed for verification. The model optimized by the optimization algorithm outperformed the normal BiGRU model (Table 10). Compared with the GRU model based on PSO optimization, the MSE, MAE, RMSE, and $R^2$ of the GRU model were improved by 19.45%, 3.38%, 10.26%, and 3.78%, respectively, and the

**Table 11 The improvement percentage of optimized evaluation model.**

| Model | Object to be compared | MSE improvement | MAE improvement | RMSE improvement | $R^2$ improvement |
|---|---|---|---|---|---|
| WOA+GRU | BiGRU | 14.86% | 2.15% | 7.72% | 2.89% |
| PSO+GRU | BiGRU | 19.45% | 3.38% | 10.26% | 2.89% |
| PSO+BiGRU | BiGRU | 19.17% | 5.07% | 10.10% | 3.73% |
| WOA+BiGRU | BiGRU | 32.09% | 9.17% | 17.59% | 6.24% |
| **MSWOA+BIGRU** | **BIGRU** | **44.98%** | **18.25%** | **25.83%** | **8.75%** |

Note:
Results for the proposed model are shown in bold.

**Table 12 Performance comparison between optimized BiGRU evaluation models.**

| MODEL | MSE | MAE | RMSE | $R^2$ |
|---|---|---|---|---|
| WOA+BiGRU | 0.001718 | 0.032079 | 0.041454 | 0.889346 |
| PSO+BiGRU | 0.002045 | 0.033526 | 0.045222 | 0.868316 |
| **MSWOA+BIGRU** | **0.001392** | **0.028871** | **0.037309** | **0.910370** |

Note:
Results for the proposed model are shown in bold.

MSE, MAE, RMSE, and $R^2$ of the GRU model based on WOA optimization were improved by 14.86%, 2.15%, 7.72%, 2.89%, and the MSE, MAE, RMSE, $R^2$ of the BiGRU model based on PSO optimization were improved by 19.17%, 5.07%, 10.10%, 3.73%, respectively. The MSE, MAE, RMSE, and $R^2$ of the BIGRU model based on WOA optimization increased by 32.09%, 9.17%, 17.59%, and 6.24%, respectively. The MSE, MAE, RMSE, and $R^2$ of the MSWOA-optimized BIGRU model increased by 44.98%, 18.25%, 25.83%, and 8.75%, respectively. Therefore, it can be concluded from experiment II that the performance of the deep learning algorithm optimized by the swarm intelligence optimization algorithm performs better than the unoptimized deep learning algorithm for cyber security situation assessment. The improvement percentage is shown in Table 11.

### Experiment III: performance comparison between optimization algorithms

The purpose of designing this part of the experiment is to test the performance of different optimization algorithms and compare them. There are currently many popular optimization algorithms. Since many network parameters need to be debugged during the training process of the neural network, it is more suitable for the swarm intelligence optimization algorithm to optimize them. MSWOA was used in our study to optimize the suggested model. The representative PSO algorithm and WOA algorithm in the swarm intelligence optimization algorithm were selected for comparison, and the performance is shown in Table 12. Comparing the BIGRU model based on MSWOA optimization with the BIGRU model based on PSO optimization, Table 13 shows that MSE, MAE, RMSE, and $R^2$ increased by 31.93%, 13.88%, 17.50%, and 4.84%, respectively. Compared to the WOA-optimized BIGRU model, MSE, MAE, RMSE, and $R^2$ of the MSWOA-optimized BIGRU model increased by 18.98%, 10.00%, 10.00%, and 2.36%, respectively. The effect of

**Table 13 The percentage of MSWOA-BiGRU the improvement degree.**

| Model | Object to be compared | MSE improvement | MAE improvement | RMSE improvement | $R^2$ improvement |
|---|---|---|---|---|---|
| MSWOA+BIGRU | PSO+BIGRU | 31.93% | 13.88% | 17.50% | 4.84% |
| MSWOA+BIGRU | WOA+BIGRU | 18.98% | 10.00% | 10.00% | 2.36% |

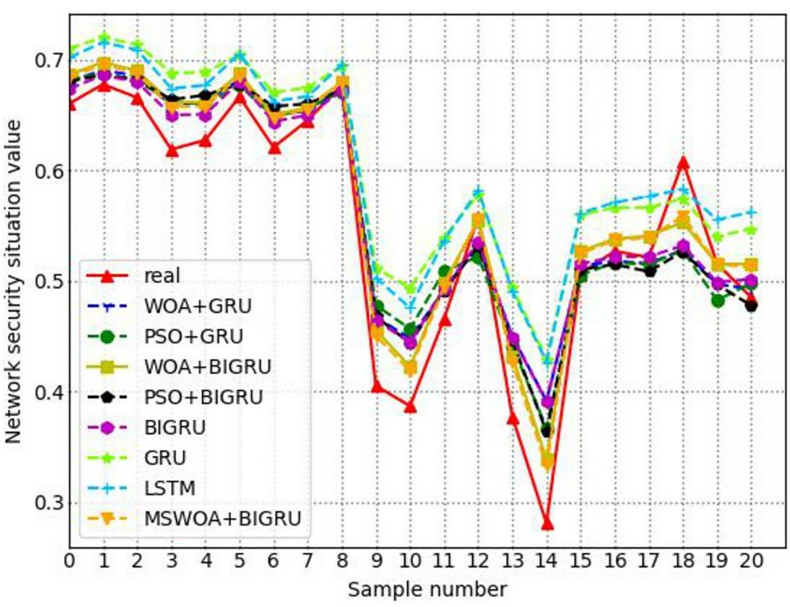

**Figure 10 Contrast of changing trends of cyber security situation values among different models.**

the MSWOA optimization evaluation model was better than the effect of the PSO and WOA optimization evaluation models.

### Experiment IV: comparison of NSSA effects

As shown in Fig. 10, the changing trend of the network security situation value calculated by the eight models is roughly the same as that of the actual network security situation value. However, a BiGRU model based on MSWOA optimization for evaluating the network security situation was proposed and the varying trend of the value was more consistent with the changing direction of the actual value. Figure 10 shows the following results:

The network security situation value calculated by the MSWOA-BiGRU model was almost in the same situation value range as the actual situation value. Except at the 18th sample point, the model result is medium, but the real situation is a high risk. The rest of the samples were all correctly evaluated.

The calculation results of the eight models at the 10th and 12th sample points were in the same range as the real situation value, but the model proposed in this article is closer to the real situation value.

**Table 14 The actual value of the network security situation and the evaluation results.**

| Sample number | MSWOA+GRU situation value | | Real situation value | |
|---|---|---|---|---|
| | The assessed value | Evaluation level | The assessed value | Evaluation level |
| 1 | 0.68620723 | High risk | 0.65993017 | High risk |
| 2 | 0.69741590 | High risk | 0.67750170 | High risk |
| 3 | 0.68966360 | High risk | 0.66551393 | High risk |
| 4 | 0.45521078 | Medium risk | 0.40492475 | Medium risk |
| 5 | 0.42207766 | Medium risk | 0.38663423 | Low risk |
| 6 | 0.49777657 | Medium risk | 0.46547872 | Medium risk |
| 7 | 0.33872858 | Low risk | 0.28058560 | Low risk |
| 8 | 0.52643970 | Medium risk | 0.50629306 | Medium risk |
| 9 | 0.53786695 | Medium risk | 0.52664030 | Medium risk |
| 10 | 0.51516360 | Medium risk | 0.51530180 | Medium risk |

To more intuitively compare the proximity between the situation value calculated by the model and the actual situation value, the evaluation results of some samples were randomly extracted from the test samples in this study. The comparison results are shown in Table 14 and the situation value calculated by the MSWOA-BiGRU model and the actual situation value are basically within the same evaluation level range. These results confirm that the proposed model in this study can accurately reflect the current network security situation.

### Experiment V: comparison with existing baselines

We compared the proposed method, MSWOA-BiGRU, with several other works related to network security situation assessment. The baseline methods used for comparison are as follows:

AE-PMU (*Tao, Liu & Yang, 2021*): Utilizing the autoencoder (AE) for data dimensionality reduction, followed by the application of the PMU deep neural network to achieve accurate and efficient NSSA.

AEDNN (*Yang et al., 2021*): Using the deep autoencoder (DAE) for data dimensionality reduction, and then employing the deep neural network (DNN) to calculate the network security status value.

GA-BP (*Wen & Wang, 2017*): Calculating the network security status value using a BP neural network optimized by GA.

BiPMU (*Liu et al., 2023*): Using bidirectional PMU (BiPMU) to evaluate the network security situation value.

PSO-WNN (*Zhao & Liu, 2018*): Using Particle Swarm Optimization (PSO) to optimize the parameters of the Wavelet Neural Network (WNN), and then applying the wavelet neural network based on particle swarm optimization to calculate the situation value.

As shown in the Table 15, compared with AE-PMU, the MSE, RMSE, MAE, and $R^2$ values increased by 63.37%, 39.50%, 55.58%, and 23.61% respectively. Compared with AEDNN, the MSE, RMSE, MAE, and $R^2$ values increased by 64.90%, 40.73%, 56.65%, and

**Table 15  Error results of different network security situation assessment methods.**

| Model | MSE | MAE | RMSE | $R^2$ |
|---|---|---|---|---|
| AE-PMU | 0.00415 | 0.04917 | 0.06445 | 0.72867 |
| AEDNN | 0.00433 | 0.05038 | 0.06578 | 0.71738 |
| GA-BP | 0.00465 | 0.05220 | 0.06820 | 0.69619 |
| BiPMU | 0.00375 | 0.04651 | 0.06122 | 0.75519 |
| PSO-WNN | 0.00268 | 0.03896 | 0.05181 | 0.82468 |
| **MSWOA-BiGRU** | **0.00152** | **0.02184** | **0.03899** | **0.90069** |

Note:
Results for the proposed model are shown in bold.

**Table 16  Results of robustness analysis under different independent experiments.**

| Metric | 1 | | 2 | | 3 | | Standard deviation of different test times | |
|---|---|---|---|---|---|---|---|---|
| | MSE | MAE | MSE | MAE | MSE | MAE | MSE | MAE |
| 0% | 0.00129 | 0.02747 | 0.00165 | 0.03111 | 0.00157 | 0.03033 | $0.00150 \pm 0.00015$ | $0.02964 \pm 0.00156$ |
| 3% | 0.00139 | 0.02857 | 0.00143 | 0.02900 | 0.00139 | 0.02857 | $0.00140 \pm 0.00002$ | $0.02871 \pm 0.00020$ |
| 5% | 0.00141 | 0.02876 | 0.00152 | 0.02992 | 0.00167 | 0.03133 | $0.00153 \pm 0.00011$ | $0.03000 \pm 0.00105$ |
| 7% | 0.00173 | 0.03183 | 0.00177 | 0.03224 | 0.00169 | 0.03148 | $0.00173 \pm 0.00003$ | $0.03185 \pm 0.00031$ |
| 10% | 0.00185 | 0.03300 | 0.00183 | 0.03282 | 0.00193 | 0.03370 | $0.00187 \pm 0.00004$ | $0.03317 \pm 0.00038$ |

25.53% respectively. Compared with GA-BP, the MSE, RMSE, MAE, and $R^2$ values increased by 67.31%, 42.83%, 58.16%, and 29.37% respectively. Compared with BiPMU, the MSE, RMSE, MAE, and $R^2$ values increased by 59.47%, 36.31%, 53.04%, and 19.27% respectively. Compared with PSO-WNN, the MSE, RMSE, MAE, and $R^2$ values increased by 43.28%, 24.74%, 43.94%, and 9.22% respectively. In our experiments, the MSWOA-BiGRU model outperformed the baseline models significantly, especially showing a noticeable improvement in situation assessment performance. Specifically, this performance boost can be primarily attributed to the BiGRU's fusion of traffic features with extracted temporal features, as well as the excellent optimization performance of MSWOA.

### Experiment VI: robustness analysis

To ensure the robustness of our experimental results, we tested the proposed method for robustness from two perspectives. Firstly, we conducted three experiments on the model. The standard deviation of the results is summarized in the Table 16, and the results in the Table 16 indicate a significantly small standard deviation. Then, to further ensure the robustness of the experimental results, we added noise data to the dataset at proportions of 1%, 3%, 5%, 7%, and 10% of the original dataset to test the model's resistance to interference. The standard deviation of the results is summarized in the Table 17. When adding 7% to 10% noise data, the model's performance experienced a noticeable decline, but the standard deviation remained small. Through robustness testing of the proposed network security posture assessment method from two perspectives and analyzing the

**Table 17 Results of robustness analysis under different proportions of noise data.**

| Metric | 0% | | 3% | | 5% | | 7% | | 10% | | Standard deviation of different noise ratios | |
|---|---|---|---|---|---|---|---|---|---|---|---|---|
| | MSE | MAE | MSE | MAE | MSE | MAE | MSE | MAE | MSE | MAE | MSE | MAE |
| 1 | 0.00129 | 0.02747 | 0.00139 | 0.02857 | 0.00141 | 0.02876 | 0.00173 | 0.03183 | 0.00185 | 0.03300 | 0.00153 ± 0.00022 | 0.02993 ± 0.00211 |
| 2 | 0.00165 | 0.03111 | 0.00143 | 0.02900 | 0.00152 | 0.02992 | 0.00177 | 0.03224 | 0.00183 | 0.03282 | 0.00164 ± 0.00015 | 0.03102 ± 0.00142 |
| 3 | 0.00157 | 0.03033 | 0.00139 | 0.02857 | 0.00167 | 0.03133 | 0.00169 | 0.03148 | 0.00193 | 0.03370 | 0.00165 ± 0.00018 | 0.03108 ± 0.00167 |

results, it is demonstrated that our model is robust, and the reported results can be considered reliable.

### Experiment VII: statistical soundness analysis

To ensure the statistical soundness of our results, we employed the Kolmogorov-Smirnov test and the Spearman coefficient for testing.

Firstly, the Kolmogorov-Smirnov test can determine whether the samples of situation assessment values and the true situation values come from the same distribution. If the two sets of samples are from the same distribution, it preliminarily suggests that the members or observations of the two sets are homogenous to some extent, or they might be influenced by similar factors or conditions. The result of the KS test is 0.071429, with a $p$-value of 0.869191. The experimental results show that if the $p$-value is less than 0.05, we do not reject the null hypothesis, concluding that the two distributions come from the same distribution.

Next, we used the Spearman coefficient to test the consistency between the samples of situation assessment values and the samples of true situation values. The Spearman coefficient is 0.920400, with a $p$-value of 3.83536E−58. If the $p$-value is less than 0.05, we reject the null hypothesis, indicating a significant Spearman correlation between the two sets of samples. Viewing the samples of posture assessment values and the true situation values as two time series, a significant Spearman correlation suggests that the two time series have a strong association. This further validates the excellent assessment performance of the method we proposed.

## CONCLUSIONS

This study primarily introduces a deep assessment model for evaluating network security situation. The model combines the improved optimization algorithm MSWOA with the deep learning model BiGRU to evaluate the current security situation of network systems. Network traffic typically exhibits temporal features. Existing methods often overlook the temporal features of traffic data and focus only on the features of traffic data at the current time point, leading to the suboptimal accuracy of existing assessment methods. The proposed approach combines the improved optimization algorithm MSWOA with the deep learning model BiGRU, effectively enhancing the situation assessment performance. By comparing its evaluation capabilities with existing network security situation assessment methods on public network security datasets, the assessment capability of this method is emphasized. The main conclusions are summarized as follows:

(1) The model proposed in this experiment has the smallest values and the least errors in MSE, MAE, and RMSE. Moreover, it has the highest $R^2$ value, indicating the best fit. Therefore, compared to other benchmark models, this model offers the best performance and more accurate network security situation assessment results.

(2) By analyzing the situation assessment performance of the model, it can be concluded that MSWOA can improve the parameters of the model. This not only enhances the model's network security situation assessment results but also endows the model with excellent robustness.

In summary, the MSWOA-BiGRU network security situation assessment model offers high accuracy and good robustness, providing effective technical support for future network security situation evaluations. Moreover, the combination of optimization algorithms and temporal neural networks can also serve as a valuable technical reference for researchers in network security situation assessment. In the future, we consider further simplifying the construction of indicators or enabling the system to automatically construct appropriate indicators. This would enhance the model's versatility, making it applicable to a broader range of network scenarios. While this article primarily focuses on numerical assessments, considering the ever-changing nature of real-world network environments, probabilistic evaluations of network security situations are more aligned with actual application scenarios. Hence, in the future, we plan to further optimize the numerical assessment into probabilistic evaluation to better account for information changes and uncertainties.

## ACKNOWLEDGEMENTS

The authors are grateful to the UNSW-NB15 public dataset, which is created by the Ixia Perfect Storm tool of the Australian network security center.

### Funding
This research was funded by Innovation Foundation Project of Gansu Provincial Department of Education, grant number 2020A-084, 2020C-29, 2021CYZC-73, 2022CYZC-57, and Supported by University-level Innovative Research Team of Gansu University of Political Science and Law. The funders had no role in study design, data collection and analysis, decision to publish, or preparation of the manuscript.

### Grant Disclosures
The following grant information was disclosed by the authors:
Gansu Provincial Department of Education: 2020A-084, 2020C-29, 2021CYZC-73, 2022CYZC-57.
University-level Innovative Research Team of Gansu University of Political Science and Law.

## Competing Interests

The authors declare that they have no competing interests.

## Author Contributions

- Shengcai Zhang conceived and designed the experiments, performed the experiments, analyzed the data, performed the computation work, prepared figures and/or tables, authored or reviewed drafts of the article, and approved the final draft.
- Qiming Fu performed the experiments, performed the computation work, prepared figures and/or tables, authored or reviewed drafts of the article, and approved the final draft.
- Dezhi An conceived and designed the experiments, authored or reviewed drafts of the article, and approved the final draft.
- Zhenxiang He analyzed the data, authored or reviewed drafts of the article, and approved the final draft.
- Zhenyu Liu analyzed the data, prepared figures and/or tables, authored or reviewed drafts of the article, and approved the final draft.

## Data Availability

The UNSW-NB15 Dataset is available from the Cyber Range Lab of UNSW Canberra and at Zenodo:

- https://research.unsw.edu.au/projects/unsw-nb15-dataset.
- Fu. (2023). UNSW-NB15 [Data set]. Zenodo. https://doi.org/10.5281/zenodo.10140548.

The code is available in the Supplemental File.

## Supplemental Information

Supplemental information for this article can be found online at http://dx.doi.org/10.7717/peerj-cs.1729#supplemental-information.

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
