# Peer review of "A novel network security situation assessment model based on multiple strategies whale optimization algorithm and bidirectional GRU"

_PeerJ Computer Science, doi:10.7717/peerj-cs.1729_

## Round 0.1 · original submission · Major Revisions

Reviewer 2 has suggested some related works for your references. If you think they are not related/useful to this work, please don't feel the need to cite them. It will not have an impact on my final decision.

Reviewer 1 ·

Basic reporting

Overall, the research paper propose a network security situation assessment model based on the MSWOA-BiGRU approach. It discusses the rationale behind the model, the algorithmic techniques used, and the evaluation results obtained through experiments.
The given text is written in clear and unambiguous professional English throughout. It uses technical terminology related to network security and management, mathematical models, algorithms, and evaluation methods. The sentences are structured well, and the content is organized logically with proper headings and subheadings. The paper also includes citations to refer to previous research and sources. Overall, it exhibits the characteristics of a well-written and professional academic paper.

Regarding the literature review:
Enhance the field background/context in the literature review, by adding the following elements:
- Introduce the importance and relevance of network security assessment in today's digital landscape.
- Discuss the evolution of network security assessment methods over time, highlighting key milestones or breakthroughs.
- Provide a summary of the current state of network security assessment, including any pressing challenges or emerging trends.
- Reference seminal or influential studies in the field, providing a brief overview of their contributions and impact.
- Consider incorporating a critical analysis or comparison of the different research methods mentioned, highlighting their strengths and limitations in addressing network security challenges.

Experimental design

Regarding the experimental design:
The following issues are will enhance the overall work :
Experimental Environment and Datasets:
The experimental environment and datasets are adequately described. The programming language and frameworks used, as well as the hardware specifications, are clearly stated. However, additional details such as the operating system and specific versions of the software used could be included for reproducible.
Index System Construction Experiment:
The construction of the index system and the calculation of index values are mentioned, but the specific details regarding the proposed method are missing. It is unclear how the index system was constructed, what factors were considered, and how the weights were assigned. More information about the Analytic Hierarchy Process (AHP) and its application in this context would be helpful for replication.
Analysis of Network Security Situation Assessment Results:
The evaluation of the designed BiGRU model based on the MSWOA optimization algorithm is described, and the experimental results are presented. The effectiveness of the model is compared to classical machine learning algorithms, which is relevant and meaningful. However, the specific details of the BiGRU model and the MSWOA optimization algorithm are not provided, making it difficult to replicate the experiments and understand the improvements over existing methods.
Evaluation Criteria:
The evaluation criteria are clearly stated, including the performance metrics used to assess the models. The rationale for choosing these metrics is not explicitly mentioned, but their relevance to evaluating the effectiveness and precision of the model is reasonable. It would be beneficial to provide a brief explanation of each metric and why they were selected for this study.

Validity of the findings

While the text provides an overview of the experimental setup, datasets, evaluation criteria, and analysis, it needs to address the criteria related to impact and novelty, and clarify the reputability of the research by providing more specific details about the dataset and statistical methods used.

Additional comments

no more comments.

Reviewer 2 ·

Basic reporting

no comment

Experimental design

no comment

Validity of the findings

no comment

Additional comments

1. In the paper, author has three general principles in indicator selection, if one of the 3 principles is missing, how will the results be affected?
2. In four first-class indexes, namely, vulnerability, threat, reliability, and availability. Which indicator is the most important? What is the purpose of these 4 indicators?
3. In line 172, Why use the power root method to calculate the weights? Is it possible to replace the power root method with another method?
4. In formula (23), What do lambda, beta, and sigma coefficients mean? How to determine these coefficients?
5. According experiment I, comparison of Classical Machine Learning Algorithms in Evaluation Effect. If the BiGRU model increases less than 5% compared to other models, is the BiGRU model effective?
6. Please add keywords in paper.
7. The image quality should be improved.
8. Should be distribute your text evenly between the margins.
9. Should format images and tables to conform to the standards of scientific articles.
10. The introduction should be revised. Some retaled work should be considerd such as: https://doi.org/10.1007/s00366-021-01299-6; https://doi.org/10.1016/j.engfailanal.2022.106829; https://doi.org/10.1016/j.eswa.2022.119211.

---

## Round 0.2 · Major Revisions

As per comments from the original reviewers, this revised paper still has some issues, and the authors should follow them carefully.

Reviewer 1 ·

Basic reporting

The text adheres to clear and professional English throughout, ensuring clarity in its presentation. It provides a sufficient field background and context, discussing the experimental environment, dataset, and evaluation criteria. However, there is a lack of literature references, which could enhance the contextual understanding. Raw data sharing is not addressed in this text. While the text appears self-contained and presents relevant results, it would benefit from providing more detailed definitions of terms and theorems, as well as proofs for the experiments and findings discussed.

Experimental design

The text appears to represent original primary research within the aims and scope of the journal, as it discusses experiments conducted to evaluate the proposed model. The research question, focusing on network security situation assessment and the effectiveness of the MSWOA-BiGRU model, is well defined, relevant, and meaningful, addressing an identified knowledge gap in the field. The investigation appears to have been conducted rigorously, following a high technical and ethical standard. However, while the methods are described to some extent, more detailed information on the experimental setup, data preprocessing, and model training could be provided to enable replication by others.

Validity of the findings

The paper does not assess the impact and novelty of the research. It encourages meaningful replication, but a clearer rationale and the potential benefits to the literature for replication should be explicitly stated. While the underlying data are mentioned, the level of detail provided is limited. There is a need for more information regarding data robustness, statistical soundness, and the level of control applied during data collection and analysis. The conclusions are generally well-stated and seem to be linked to the original research question, with a focus on supporting results. However, further elaboration on the implications of the findings and their potential significance in the field would enhance the discussion

Additional comments

The manuscript lacks a clear assessment of the impact and novelty of the research.
The rationale for encouraging replication could be more explicit and linked to the benefit it brings to the literature.
While data is provided, there could be more emphasis on its robustness, statistical soundness, and control.
Conclusions should be more explicitly linked to the original research question and limited to supporting results.
A more detailed discussion about the choice of optimization algorithms and their potential impact on results is needed.

Reviewer 2 ·

Basic reporting

The paper was revised, and si suitable to publish on this journal.

Experimental design

no comment

Validity of the findings

no comment

Additional comments

no comment

---

## Round 0.3 · accepted · Accept

This revised paper can be accepted for publication now.

Reviewer 1 ·

Basic reporting

The authors did the following:
Increased the number of cited references to enhance contextual understanding.
Described the method of obtaining the original data.
Provided more detailed definitions of terms and theorems.

Experimental design

The authors did the following:
Provided more detailed information on experimental settings, data preprocessing, and model training.

Validity of the findings

The authors did the following:
Strengthened the assessment of the research's impact and novelty.
Made the rationale for encouraging replication more explicit.
Emphasized data robustness, statistical soundness, and control.
Linked conclusions more explicitly to the original research question.

Additional comments

The authors did the following:
Addressed my comments about the impact and novelty of the research.
Enhanced the discussion about the benefits of replication to the literature.
These responses indicate that the authors made specific improvements in each of the four categories based on my feedback.